

# Carbon sequestration along a gradient of tidal marsh degradation in response to sea level rise

Mona Huyzentruyt[1], Maarten Wens[1], Gregory Fivash[1], David Walters[2], Steven Bouillon[3], Joel Carr[2], Glenn Guntenspergen[4], Matt Kirwan[5], Stijn Temmerman[1]

[1] ECOSPHERE Research Group, University of Antwerp, Antwerp, Belgium

[2] U.S. Geological Survey, Eastern Ecological Science Center, Laurel, MD, USA

[3] Department of Earth and Environmental Sciences, KU Leuven, Leuven, Belgium

[4] U.S. Geological Survey, Eastern Ecological Science Center, Duluth, MN, USA

[5] Virginia Institute of Marine Science, William & Mary, Gloucester Point, Virginia

*Correspondence to*: Mona Huyzentruyt (mona.huyzentruyt@uantwerpen.be)

**Abstract.**

Tidal marshes are considered one of the world's most efficient ecosystems for belowground organic carbon sequestration and hence climate mitigation. Marsh systems are however also vulnerable to

degradation due to climate-induced sea level rise, whereby marsh vegetation conversion to open water often follows distinct spatial patterns: levees (i.e. marsh zones <10 m from tidal creeks) show lower vulnerability of vegetation conversion to open water than basins (i.e. interior marsh zones >30 m from creeks).  Here, we use sediment cores to investigate spatial variations in organic carbon accumulation rates (OCAR) in a microtidal system (Blackwater marshes, Maryland, USA): (1) across a gradient of

marsh zones with increasing marsh degradation, assessed as increasing ratio of unvegetated versus vegetated marsh area and (2) by comparing levees versus basins. We show that OCAR is up to four times higher on marsh levees than in adjacent basins. The data suggest that this is caused by spatial variation in three processes: sediment accretion rate, vegetation productivity, and sediment compaction, which are all higher on levees. Additionally, OCAR was observed to increase with increasing degree of marsh

degradation in response to sea level rise. We hypothesize this may be due to more soil waterlogging in more degraded marsh zones, which may decrease carbon decomposition. Our results highlight that tidal marsh levees, in a microtidal system, are among the fastest soil organic carbon sequestration systems on





Earth, and that both levees and basins sustain their carbon accumulation rate along gradients of increasing

marsh degradation in response to sea level rise.



**Graphical abstract**

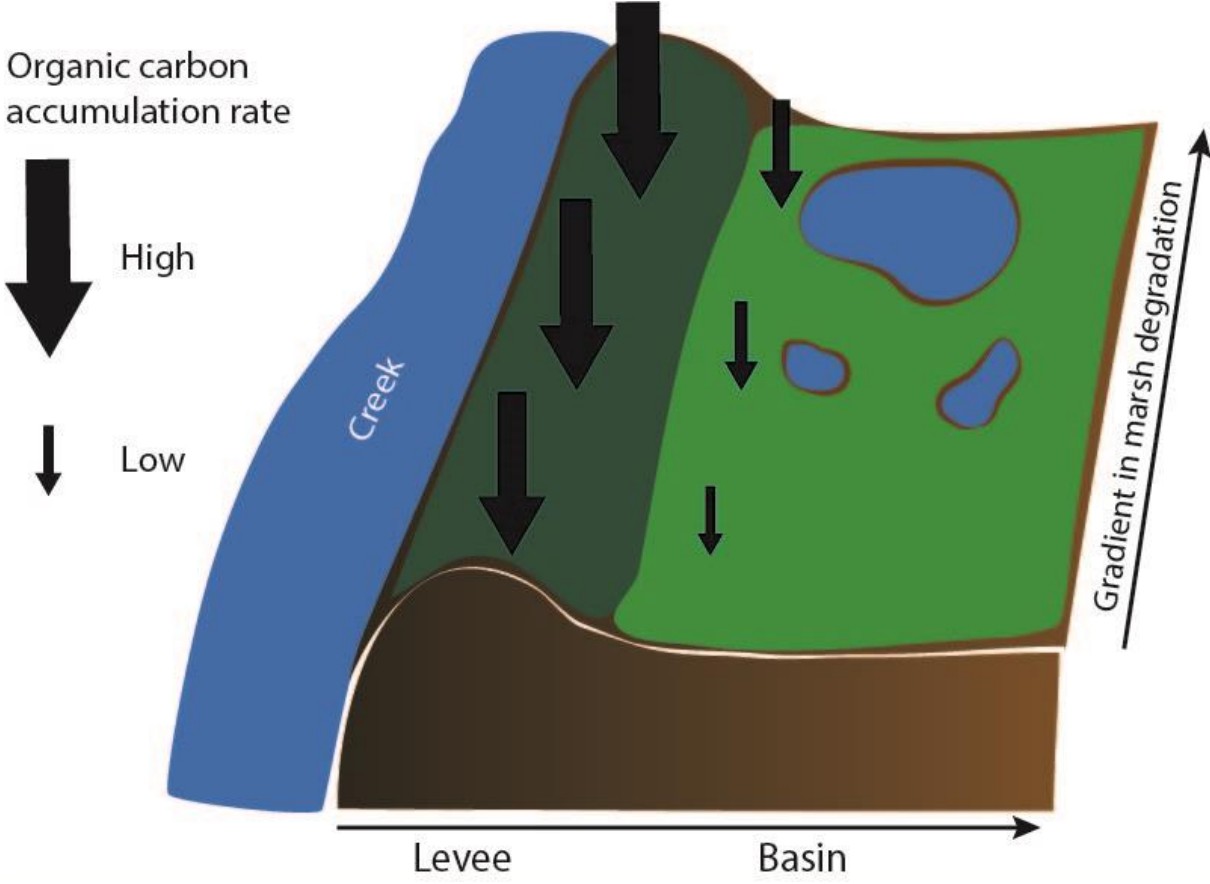

## 1 Introduction

Tidal marsh ecosystems are among the most efficient ecosystems on Earth in terms of long-term carbon

sequestration per surface area, with an average organic carbon accumulation rate (OCAR) of 250 g m$^{-2}$ y$^{-1}$ and high-end values up to 1800 g m$^{-2}$ y$^{-1}$ (Huyzentruyt et al., 2024; Temmink et al., 2022). This efficiency stems from the fact that the organic carbon can originate from two main sources: (1) locally produced carbon by highly productive marsh vegetation and (2) externally derived (e.g. terrestrial or marine) carbon supplied as suspended matter in the water and deposited by tidal inundation (McLeod et

al., 2011; Middelburg et al., 1997; Williamson et al., 2025). Additionally, due to tidal inundation, the marsh sediment bed is waterlogged for a large part of the tidal cycle, reducing the amount of oxygen





available for carbon decomposition (Luo et al., 2019). A final reason for their high carbon sequestration efficiency is that as more sediment is accreted on the marsh surface, the previous layers get buried and reach more anoxic soil environments, which slows down the mineralisation of the present soil organic

carbon (Kirwan & Mudd, 2012; Müller & Suess, 1979; Van De Broek et al., 2016).

The long-term stability of tidal marsh systems is impacted by anthropogenic and natural processes, and how this affects their carbon sequestration efficiency remains relatively poorly studied. One of the major threats that these systems are facing is climate-change induced sea level rise (Craft et al., 2009; Day et al., 2024; Kirwan & Megonigal, 2013; Morris et al., 2002; Schuerch et al., 2018). On the one hand certain

marshes can keep up with sea level rise, due to positive feedbacks between tidal inundation duration, sediment accretion, and surface elevation gain, in particular macro-tidal marshes with high sediment supply (Kirwan et al., 2016). For such marsh sites previous studies have found an increase in organic carbon accumulation rate with increasing sea level rise rate, due to the positive feedback between tidal inundation duration, sediment accretion rate and hence organic carbon accumulation rate (Herbert et al.,

2021; Huyzentruyt et al., 2024; Suello et al., 2025; Wang et al., 2021). On the other hand, there are marsh sites where sediment accretion rates cannot keep up with the local relative sea level rise rate, which is a particular risk in micro-tidal marshes with limited sediment supply and high rates of relative sea level rise (Kirwan et al., 2016). This is for instance the case in the Chesapeake Bay (Ganju et al., 2013; Kearney et al., 1988; Qi et al., 2021; Schepers et al., 2017), the Mississippi River delta (DeLaune & White, 2012;

Herbert et al., 2021; Ortiz et al., 2017) and the Venice Lagoon (Fagherazzi et al., 2006). Within these systems, certain marsh zones are experiencing sediment accretion rates that are too low to keep up with sea level rise, resulting in increasing tidal inundation stress on marsh vegetation, reduced vegetation productivity and eventually vegetation die-off. The resulting bare soil patches or shallow ponds that form inside marshes, and their surface area relative to the surrounding remaining vegetated marsh area (so-

called unvegetated-vegetated ratio, UVVR), is considered here a proxy for the state or degree of marsh degradation, in line with previous studies (Ganju et al., 2017). An important question is how this degree of marsh degradation (measured as UVVR) in response to sea level rise affects the organic carbon sequestration efficiency in the remaining vegetated marsh zones.



Marshes with a sediment accretion deficit lose elevation relative to the rising sea level and hence experience increasing tidal inundation duration, which will likely affect organic carbon sequestration (Morris et al., 2002; Mudd et al., 2009). Increased inundation duration is likely to lead to a decrease in available oxygen in the sediment and an increase in the build-up of phytotoxins, such as sulphides, in the sediment (Himmelstein et al., 2021; Linthurst, 1979; Mendelssohn & Mckee, 1988), both of which negatively affect vegetation growth. The relationship between vegetation productivity and inundation duration varies between different species (Janousek et al., 2016; Kirwan & Guntenspergen, 2015; Snedden et al., 2015; Watson et al., 2014). Some species, such as *Schoenoplectus americanus* show a parabolic relation, with a maximal biomass productivity for an intermediate inundation duration (Kirwan & Guntenspergen, 2015; Langley et al., 2013). Other species such as *Spartina patens* and *Spartina alterniflora* show a decrease in biomass and productivity with increased inundation duration (Janousek et al., 2016; Kirwan & Guntenspergen, 2015; Langley et al., 2013; Snedden et al., 2015; Watson et al., 2014). A decrease of vegetation productivity with increasing inundation duration could potentially result in a lower OCAR in the remaining marsh, because of lower organic inputs and lower trapping of external sediment. However, increased inundation duration is also expected to result in lower aerobic microbial mineralization of the extant sediment organic carbon, which could result in higher OCAR rates in the remaining marsh. Further, increased inundation duration may induce to some extent increased supply and deposition of external sediment and organic carbon. Hence, it is difficult to predict what the overall response is of OCAR to increased tidal marsh inundation, where sea level rise rate is higher than sediment accretion rate.

The degradation of marsh vegetation in response to sea level rise is typically not a spatially uniform process (Schepers et al., 2017), so we may also expect that changes in OCAR in response to a different degree of marsh degradation will follow distinct spatial patterns within marshes. Reduced vegetation productivity and vegetation die-off in response to increased tidal inundation especially occurs in low elevation, sediment starved interior marsh basins, located further away from tidal channels, whereas marsh zones bordering tidal channels often show lower vulnerability to reduced vegetation productivity and die-off (Kearney et al., 1988; Luk et al., 2023; Schepers et al., 2017). This spatio-temporal pattern of reduced vegetation productivity and die-off is shown to be related to the typical micro-topographical



gradient that forms in tidal marshes (Schepers et al., 2017), with higher elevated levees close to (<10-20 m from) tidal channels and lower elevated basins further away from channels (~20-100 m). Levees can be typically 10-40 cm higher than the basins (Christiansen et al., 2000; Redfield, 1972a; Temmerman et al., 2003). This micro-topographical gradient is formed by higher sedimentation rates close to the creeks (French et al., 1995; Reed et al., 1999; Temmerman et al., 2003) and results in differences in hydrological (Ursino et al., 2004; Van Putte et al., 2020) and biogeochemical (Kostka et al., 2002) processes between levees and basins. Levees are known to have more sediment pore water drainage and thus higher sediment oxygen levels (Ursino et al., 2004; Van Putte et al., 2020), associated with higher vegetation productivity compared to basins (Gleason & Zieman, 1981; Linthurst, 1979; Mendelssohn, 1981). Further, research has shown that there is a difference in microbial decomposition between levee and basin sites, with higher rates of decomposition occurring on levees and to deeper depths compared to basin sites (Kostka et al., 2002). It can be expected that these differences between basins and levees can also lead to differences in OCAR. However, there are currently no studies that have investigated the dynamics of OCAR along levee-basin gradients in marsh zones with a different degree of marsh degradation in response to sea level rise, which hampers our ability to predict the long-term stability of carbon in these systems as they progressively degrade in response to sea level rise.

With this study we aim to quantify OCAR in vegetated marsh zones along two spatial gradients reflecting changing environmental conditions: (1) a gradient in marsh degradation (UVVR) and (2) a gradient from levees to basins. The Blackwater marshes in Maryland, USA, provide a unique opportunity to address these open questions. Schepers et al. (2017) found that the spatial gradient in marsh degradation observed within this marsh complex can be considered a chronosequence of increasing marsh degradation in response to sea level rise. In this study, we utilize this gradient to investigate how OCAR varies: (1) across marsh zones with increasing degree of degradation (increasing UVVR), and (2) within marsh zones, across the micro-topographic gradient from levees to basins.



## 2 Materials and methods

### 2.1 Study area

The Blackwater marshes are located along the Blackwater and Transquaking rivers (Fig. 1), which discharge into the Fishing Bay, a tributary embayment of the Chesapeake Bay (Maryland, USA, Fig. 1).

These marshes are organogenic and micro-tidal, with a spring tidal range varying between less than 0.2 m upstream to over 1.0 m at the Fishing Bay (Ganju et al., 2013). In the marshes a mixture of mesohaline, intertidal vegetation can be found, including *Spartina cynosuroides* (L.) Roth on the levees, forming a belt of ca. 10-20 m wide adjacent to channels, and patches of *Schoenoplectus americanus* (Pers.) and a mixture of *Spartina alterniflora* Loisel and *Spartina patens* Roth in the basins, at more than 10-20 m from

channels. In this system, the measured difference in surface elevation between levees and basins is usually 0.07-0.17 m (Table 1). The part of the Chesapeake Bay closest to the Blackwater marshes experiences a relative sea level rise rate of 4.06 mm y$^{-1}$ (measured between 1943 and 2024; NOAA station Cambridge, MD, 8571892, https://tidesandcurrents.noaa.gov/sltrends/, accessed on 6/30/2025), which is higher than the average historical sediment accretion rate of 3.9 mm y$^{-1}$ measured in the Blackwater marshes (Ganju

et al., 2013). This accretion deficit has led to a spatial gradient in marsh degradation, with stable marshes close to the Fishing Bay and increasing historical conversion of marsh to ponds moving upstream along the Blackwater River (Schepers et al., 2017).





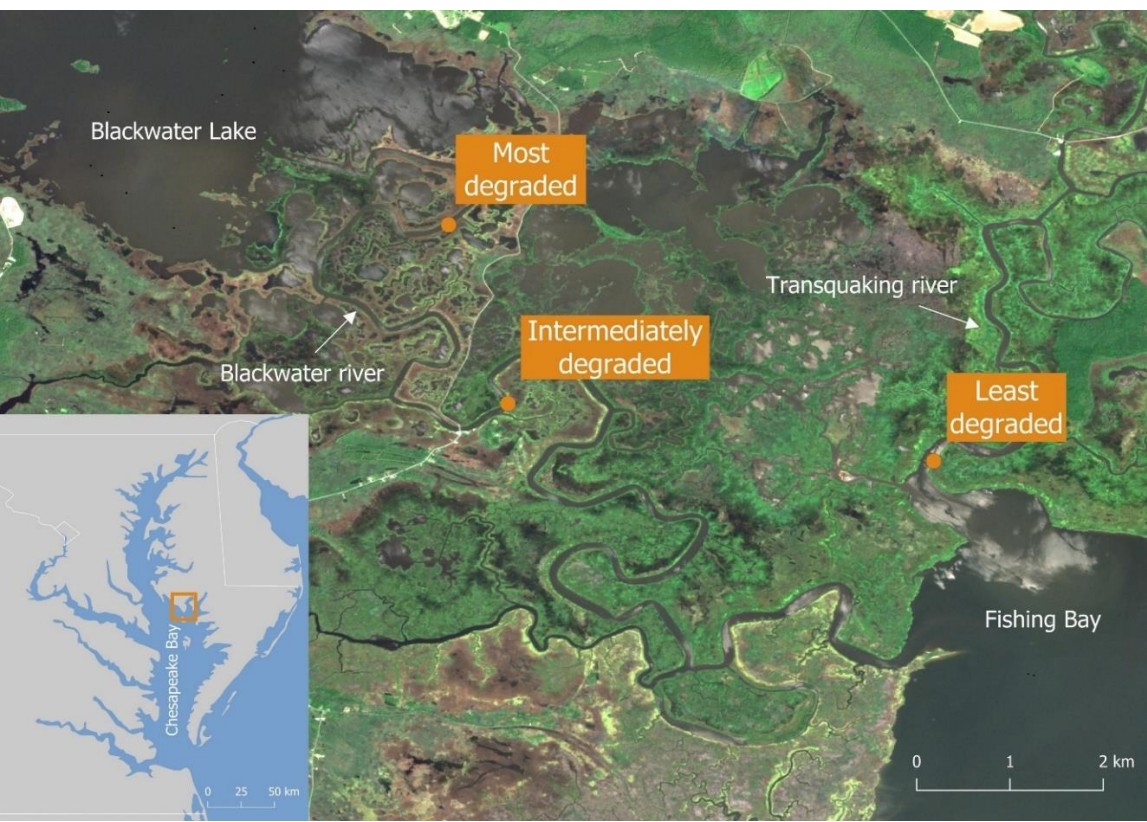

**Figure 1: Map showing the location of (left) the Blackwater marshes in the Chesapeake Bay and (right) the**
**location of the selected zones along the marsh degradation gradient (Copernicus – Sentinel data [2025].**
**Retrieved from Google Earth Engine, processed by ESA).**

## 2.2 Fieldwork setup

Three marsh zones were selected along the marsh degradation gradient, referred to as (1) least degraded,
(2) intermediately degraded and (3) most degraded (Fig. 1). This gradient is characterized by an increasing
unvegetated-vegetated ratio (UVVR; Ganju et al., 2017) moving from the least to most degraded zone
(Fig. S1, Table 1). Within each zone, samples were collected on levee and on basin locations. Because
the basins of the least degraded and intermediately degraded zone contained distinct patches of two
vegetation types, samples were taken within these zones at two basin locations, i.e. in each of the two
vegetation types (one dominated by *S. americanus* and the other by a mixture of *S. alterniflora* and *S.*
*patens*), but only one levee location was sampled (dominated by *S. cynosuroides*). In the most degraded
zone, one levee location (dominated by *S. cynosuroides*) and one basin location (dominated by *S.*





*americanus*) were sampled, because no *S. alterniflora* and *S. patens* community was present here. This resulted in a total of 8 sampling locations (Table 1). At each sampling location four replicate cores were taken, of which one was used for radiometric dating (refer to section 2.3.2) and three were used for bulk

density and organic carbon analysis.

**Table 1: Overview of the different sampling sites and their corresponding vegetation, elevation with relation to the North American Vertical Datum of 1988 (NAVD88), shortest distance to the edge of the channel, the mean high and low water level with relation to the NAVD88, the hydroperiod (%), and the unvegetated-vegetated ratio (UVVR). Surface elevations were measured at each site using Real-Time Network (RTN)**

**surveys, collected with a Trimble R10 GNSS receiver with cm-level accuracy. The hydroperiod is the average proportion of time that the marsh surface is inundated in each tidal cycle. The UVVR was calculated within a 200 m region in each degradation zone (more information in Supplementary Information).**

| Position along the marsh degradation gradient | Position along the micro-topographical gradient | Vegetation type | Elevation (m NAVD88) | Shortest distance to channel (m) | Mean high water level (m NAVD88) | Mean low water level (m NAVD88) | Hydroperiod (%) | UVVR |
|---|---|---|---|---|---|---|---|---|
| Least degraded | Levee | *Spartina cynosuroides* | 0.60 | 7.59 | 0.476 | -0.167 | 2.51 | 0 |
| Least degraded | Basin | *Spartina alterniflora* | 0.43 | 42.10 | 0.476 | -0.167 | 15.49 | 0 |
| Least degraded | Basin | *Schoenoplectus americanus* | 0.52 | 58.53 | 0.476 | -0.167 | 5.55 | 0 |
| Intermediately degraded | Levee | *Spartina cynosuroides* | 0.32 | 3.98 | 0.229 | 0.025 | 3.67 | 0.016 |
| Intermediately degraded | Basin | *Spartina alterniflora* | 0.18 | 37.78 | 0.229 | 0.025 | 31.28 | 0.016 |
| Intermediately degraded | Basin | *Schoenoplectus americanus* | 0.23 | 57.45 | 0.229 | 0.025 | 13.48 | 0.016 |
| Most degraded | Levee | *Spartina cynosuroides* | 0.22 | 9.71 | 0.183 | 0.123 | 14.34 | 0.143 |
| Most degraded | Basin | *Schoenoplectus americanus* | 0.15 | 55.32 | 0.183 | 0.123 | 48.99 | 0.143 |

## 2.3 Sample collection and analysis

Prior to sediment sampling, above ground vegetation biomass was clipped from a 25x25 cm surface area, transported to the lab and stored cool prior to drying. Sediment samples were collected by vertically pushing down a metal coring tube with a diameter of 10 cm and length of 60 cm and a razor blade at the



bottom to cut through the below-ground roots and plant structures. Before extraction of the tube, the inner and outer length of the tube were measured (Fig. S2), in order to calculate the total rate of compaction of the sediment core inside the tube. After core extraction, each core was transported to the lab, where it was frozen. After freezing, the cores were sliced at intervals of approximately 1 cm, their exact thickness was measured, and they were stored in the freezer until drying. The sediment and vegetation samples were dried at 55°C for at least 48 hours. For the sediment samples, every other depth was used for further analysis.

### 2.3.1 Bulk density

The volume of each sediment sample was calculated based on the diameter of the tube and the measured thickness of each slice, and was corrected for the measured compaction during coring, to obtain an estimate of in situ sediment volume. After drying, the samples were weighed and the bulk density was calculated by dividing the weight by the volume.

### 2.3.2 Sediment accretion rates

The sediment accretion rates were calculated using radiometric dating, which was done on one replicate core for each location. The dried sediment was finely ground and tightly packed into a pre-weighed petri dish of known volume. The petri dishes were sealed using vinyl electrical tape and paraffin wax and left to rest for a minimum of 30 days, to establish an equilibrium between the radionuclides ($^{226}$Ra and daughter products $^{214}$Pb and $^{214}$Bi). After the resting period, each sample was analysed for $^{210}$Pb (46.5 keV photopeak), $^{214}$Pb (295, 352 keV photopeaks), and $^{214}$Bi (609 keV photopeak) activity by gamma spectroscopy using shielded ultra-low background Canberra GL 2020 Low Energy Germanium (LEGe) for periods of 24 h (FitzGerald et al., 2021). The values are corrected for background noise and adjusted for sample depth attenuation and detector sensitivity. After these adjustments, the concentration of total $^{210}$Pb, supported $^{210}$Pb, which is derived from the decay of the naturally occurring $^{226}$Ra, and excess $^{210}$Pb$_{xs}$ was computed. For each sample, the difference between the total and supported $^{210}$Pb is calculated as the atmospherically deposited excess $^{210}$Pb$_{xs}$. The accretion rates were calculated from the $^{210}$Pb rates with a Constant Flux-Constant Sedimentation (CF-CS; Krishnaswamy et al., 1971) model, where a constant rate





of sediment accumulation and a constant $^{210}$Pb flux is assumed. The accretion rate is calculated from the slope of the linear regression line between the natural log of the $^{210}$Pb$_{xs}$ activity against sample depth (Eq. 1 and refer to Supporting Information S3 for more information).

$$S \ (mm \ y^{-1}) = \frac{\lambda}{m} * 10 \ , \tag{1}$$

Where $\lambda$ is the decay constant for $^{210}$Pb (0.03101 y$^{-1}$) and m is the slope of the previously mentioned regression.

### 2.3.3 Suspended sediment

Water samples were collected at one location along the Blackwater River and stored in the fridge until further analysis. The water samples were filtered using pre-weighed, pre-baked (for 4-5h at 450°C) glass microfiber paper filters (0.7 µm pore size, GE Bio-Sciences 1825-047). After filtration, the filters were dried at 55°C and stored in petri dishes. Afterwards, the filters were acidified with HCl fumigation to remove carbonates.

### 2.3.4 Organic carbon content and sources of carbon

The three remaining sediment cores obtained for each sampling location were used for organic carbon analysis. To determine the organic carbon content of the sediment samples, the dried sediment was first ground finely. The organic carbon content and $\delta^{13}$C of the samples was determined using EA-IRMS (Thermo EA 1110 coupled to a Thermo Delta V Advantage isotope ratio mass spectrometer), after acidification of the samples to exclude inorganic carbonates. The calibration of the EA-IRMS was done using three different standards. First the IAEA-600 (caffeine), which is a certified standard distributed by the International Atomic Energy Agency. In addition to caffeine, Leucine and Tuna are used as in-house standards of the laboratory and are calibrated against certified standards (IAEA-600, IAEA-N1, IAEA-CH-6). The measured $\delta^{13}$C values are expressed relative to the international standard VPDB (Vienna PeeDee Belemnite) and show an analytical uncertainty of 0.15 ‰ or better.



The $\delta^{13}C$ values were also measured for above-ground vegetation, by analysing finely ground vegetation samples, and for suspended sediment samples. For the analysis of the suspended sediment samples, the filters were cut into four equal parts and one part was used for the $\delta^{13}C$ analysis. Additional blank filters were pre-baked and used to blank-correct the $\delta^{13}C$ data. Aboveground vegetation and suspended sediment are seen as the potential sources of autochthonous versus allochthonous carbon, respectively. The $\delta^{13}C$ values of the sediment samples were compared with these values to estimate the contribution of autochthonous versus allochthonous sources to the organic carbon preserved in the sediment.

### 2.3.5 Organic carbon density and accumulation rate

From the organic carbon content (OC; %) and the bulk density (BD; g cm$^{-3}$), the organic carbon density (OCD; g cm$^{-3}$) is calculated (Eq. 2)

$$OCD = (OC * BD)/100 , \tag{2}$$

The organic carbon accumulation rate (OCAR; g m$^{-2}$y$^{-1}$) is calculated from the organic carbon density (OCD) and the sediment accretion rate (SAR; mm y$^{-1}$; Eq.3).

$$OCAR = OCD * SAR * 1000 , \tag{3}$$

### 2.4 Statistical analysis

For sediment accretion rates and bulk density, the difference between levee and basin locations with *Schoenoplectus* and *Spartina* was investigated using ANOVA in R version 4.4.1 (R Core Team, 2022). For the organic carbon content, density and accumulation rate, the average value for each core was used to minimize the effect of the depth profiles. The difference between levee and basin locations and zones with a different degree of marsh degradation were also investigated using ANOVA.

## 3 Results

### 3.1 Bulk density and sediment accretion

Bulk density was significantly higher ($p < 0.05$) in the levee locations (0.34 g cm$^{-3}$; Fig. 2A) compared to the basin locations with *Spartina* (0.14 g cm$^{-3}$) and with *Schoenoplectus* (0.127 g cm$^{-3}$). Sediment




accretion rates (more information in Supporting Information S3) were significantly different (p < 0.05) between the levees (10.88 mm y⁻¹; Fig. 2B) and the basins with *Schoenoplectus* (3.83 mm y⁻¹) and basins with *Spartina* (3.56 mm y⁻¹).

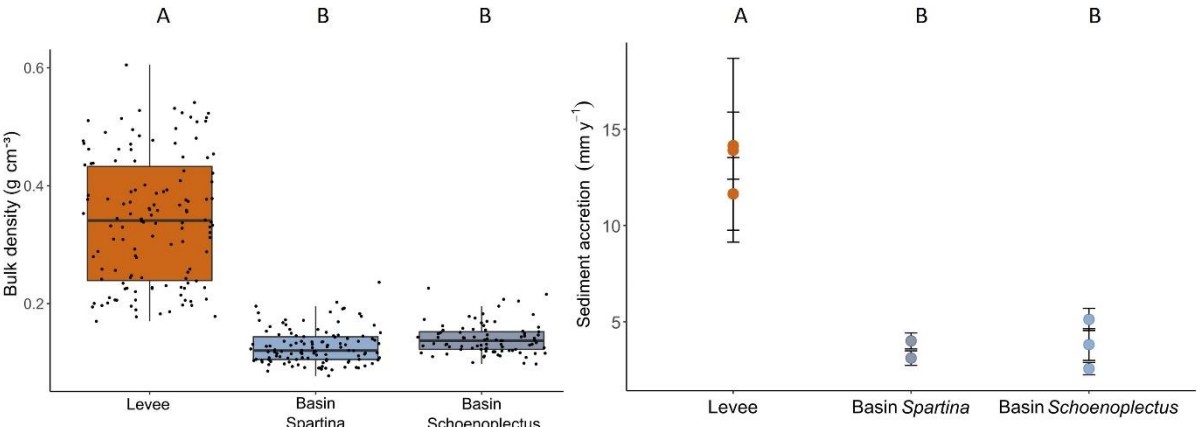

**Figure 2: Dry bulk density (left) and sediment accretion rates (right) determined with radiometric dating along the levee basin gradient. The data shown are pooled for the least degraded, intermediately degraded and most degraded zone. The letters above indicate the significance of the differences, where observations with the same letters are not significantly different from each other.**

## 3.2 Organic carbon content, density and accumulation rate

The organic carbon content (%; OC) along the marsh degradation gradient at every zone was significantly lower (p < 0.05) on the levees than in the basins with *Schoenoplectus* and *Spartina*. The only exception was the intermediately degraded zone, where the basin location with *Spartina* (50.7%) was not significantly different from the levee location (44.2%). In the intermediately and least degraded zones, there was no significant difference (p > 0.05) between the basin with *Spartina* (50.7% and 47.7%, respectively) and the basin with *Schoenoplectus* (55.9% and 42.6%, respectively), but the basin with *Schoenoplectus* in the intermediately degraded zone had a significantly higher (p < 0.05) OC than the basin with *Schoenoplectus* in the least degraded zone. The OC in the basin with *Schoenoplectus* of the most degraded zone (62.55%) was significantly higher (p > 0.05) than in both basins in the least degraded zone and the basin with *Spartina* in the intermediately zone (Fig. 3).




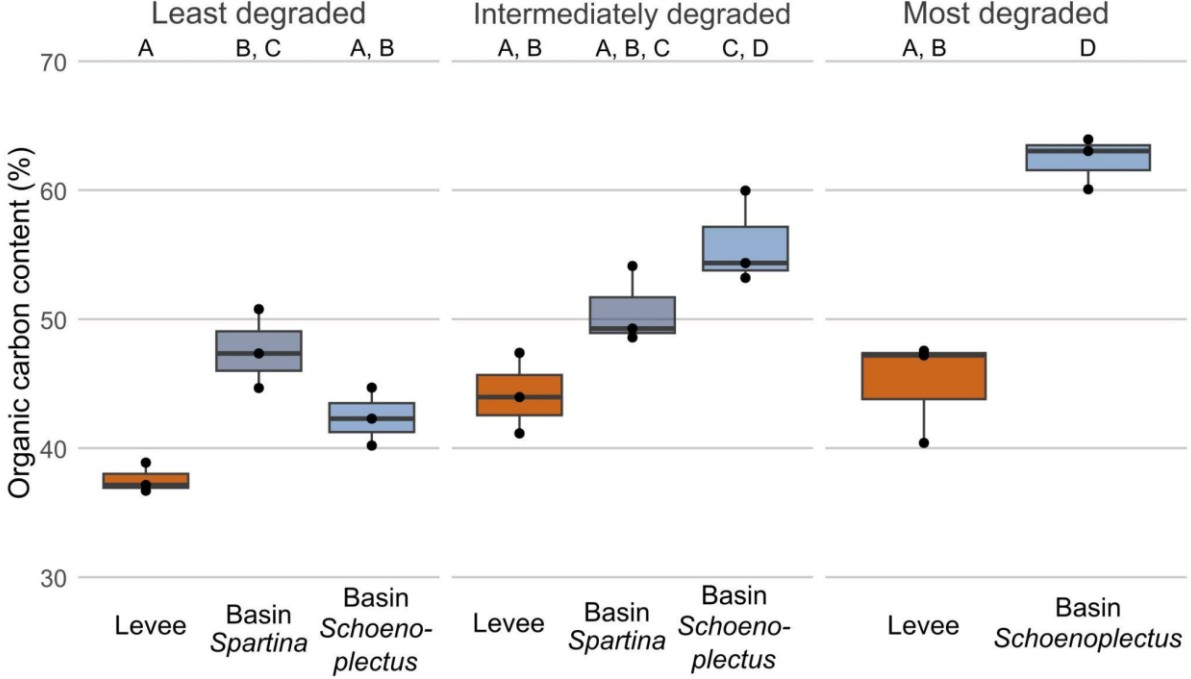

260

**Figure 3: Organic carbon content along the marsh degradation and levee-basin gradients. The colours correspond to the sampling location along the levee-basin gradient. The letters above indicate the significance of the differences, based on ANOVA results, where observations with the same letters are not significantly different from each other.**

For organic carbon densities (g cm$^{-3}$; OCD) the values were significantly higher ($p < 0.05$) on the levees compared to the basins, for all zones along the marsh degradation gradient. The values in the basins also increased along the degradation gradient, with the highest basin values found in the most degraded zones (0.030 g cm$^{-3}$), followed by the intermediately degraded basin with *Schoenoplectus* (0.029 g cm$^{-3}$) and *Spartina* (0.028 g cm$^{-3}$). The lowest values were found in the least degraded basin with *Schoenoplectus* (0.025 g cm$^{-3}$) and with *Spartina* (0.024 g cm$^{-3}$). There was however no significant difference between the values in the different basin zones (Fig. 4).





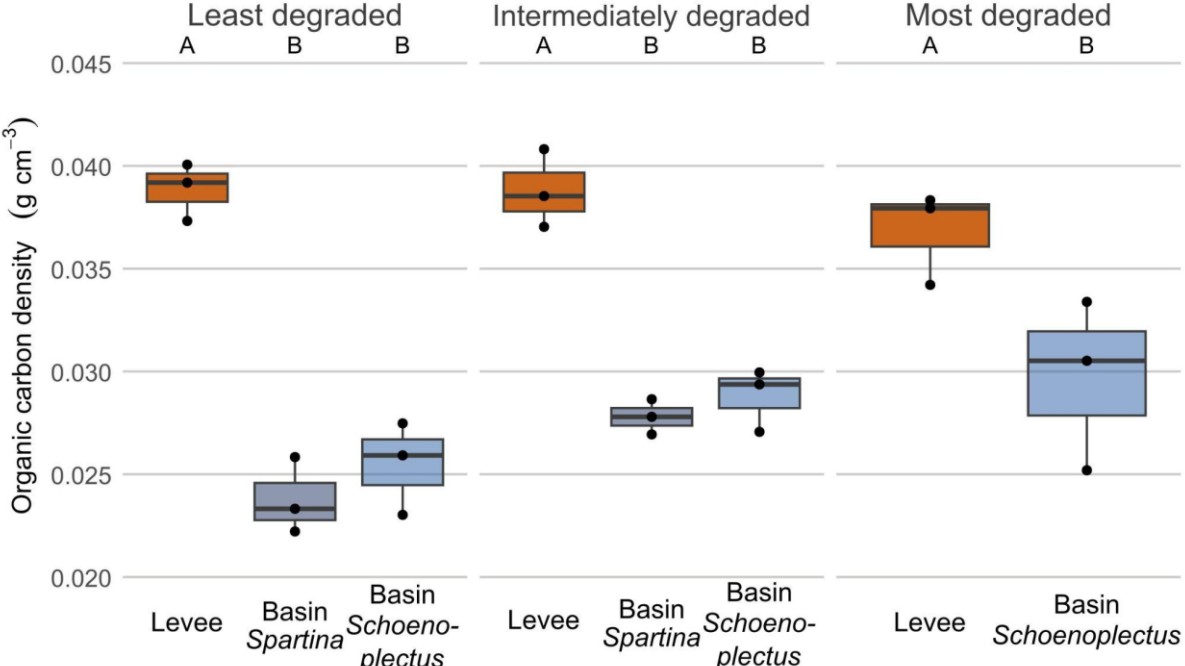

**Figure 4: Organic carbon density along the degradation and levee-basin gradient. The colours correspond to the sampling location along the levee-basin gradient. The letters above indicate the significance of the differences, based on ANOVA results, where observations with the same letters are not significantly different from each other.**

The organic carbon accumulation rate (g $m^{-2}y^-1$; OCAR) was significantly higher (p < 0.05) on the levees compared to the basins in all zones. When looking at levees only, OCAR was significantly lower in the least degraded zone (452.4 g $m^{-2}y^{-1}$) compared to the intermediately (539.6 g $m^{-2}y^{-1}$) and most degraded (521.1 g $m^{-2}y^{-1}$) zones. The OCAR in the basin with *Schoenoplectus* at the most degraded zone (152.1 g $m^{-2}y^{-1}$) was significantly higher (p < 0.05) than in the basins with *Schoenoplectus* at the intermediately (73.7 g $m^{-2}y^{-1}$) and least degraded (97.2 g $m^{-2}y^{-1}$) zones. Within the least and intermediately degraded zone, there was no difference between the basin with *Schoenoplectus* (97.2 g $m^{-2}y^{-1}$ and 73.7 g $m^{-2}y^{-1}$ resp.) and the basin with *Spartina* (95.3 g $m^{-2}y^{-1}$ and 86.4 g $m^{-2}y^{-1}$ resp.; Fig. 5).





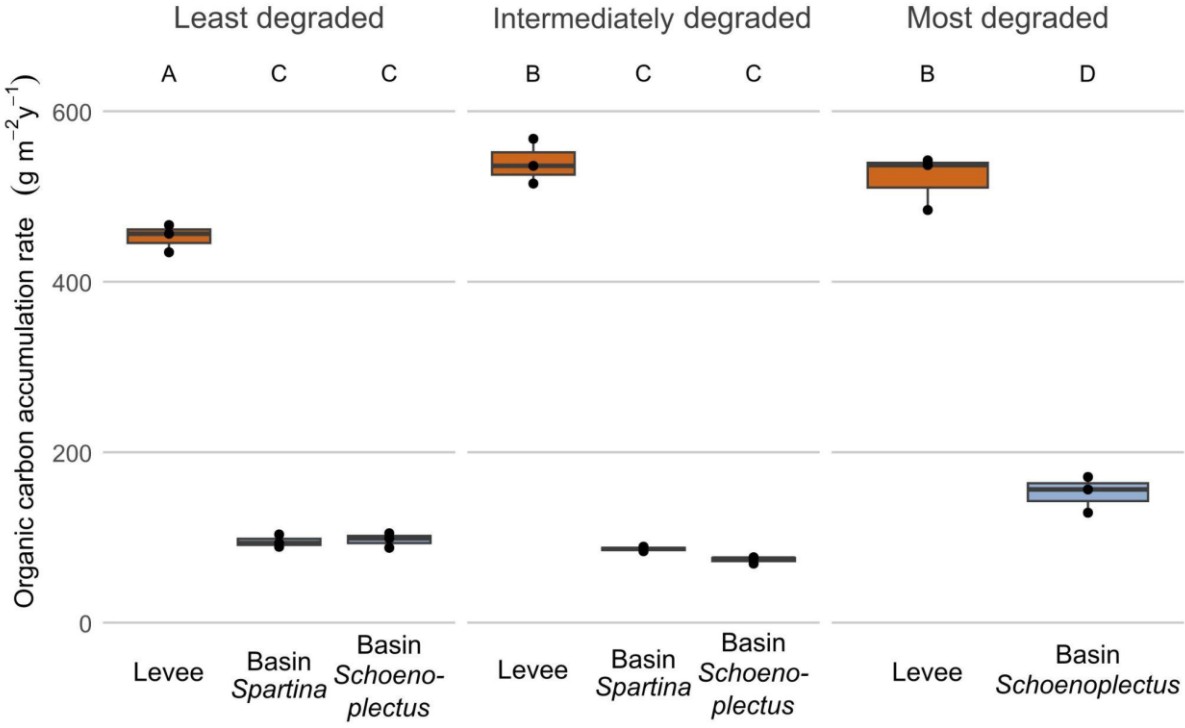

**Figure 5: Organic carbon accumulation rate along the degradation and levee-basin gradient. The colours correspond to the sampling location along the levee-basin gradient. The letters above indicate the significance of the differences, based on ANOVA results, where observations with the same letters are not significantly different from each other.**

**3.3 Sources of carbon**

For the basin locations with a mix of *Spartina alterniflora* and *S. patens* vegetation, the average $\delta^{13}$C value of the sediment OC (SOC) was approximately -16‰ for the least and the intermediately degraded zone, which is close to the $\delta^{13}$C value of the C4 *Spartina* vegetation of both the levee and the basin with *Spartina* (-14.4‰). For the levee locations, which are dominated by *Spartina cynosuroides* vegetation,

the average SOC $\delta^{13}$C value was approximately -21‰, which is between the C4 (-14.4‰) and suspended sediment (-26.3‰) $\delta^{13}$C values. For the basin with *Schoenoplectus*, there was a high variation in $\delta^{13}$C values in the least degraded and intermediate zones, ranging from SOC $\delta^{13}$C values close to the C4 vegetation values to values closer to the C3 *Schoenoplectus* vegetation value (-24.9‰). In the basin with




*Schoenoplectus* at the most degraded zones, the average SOC $\delta^{13}$C value (-24.4‰) corresponded to that

of the C3 vegetation (Fig. 6).

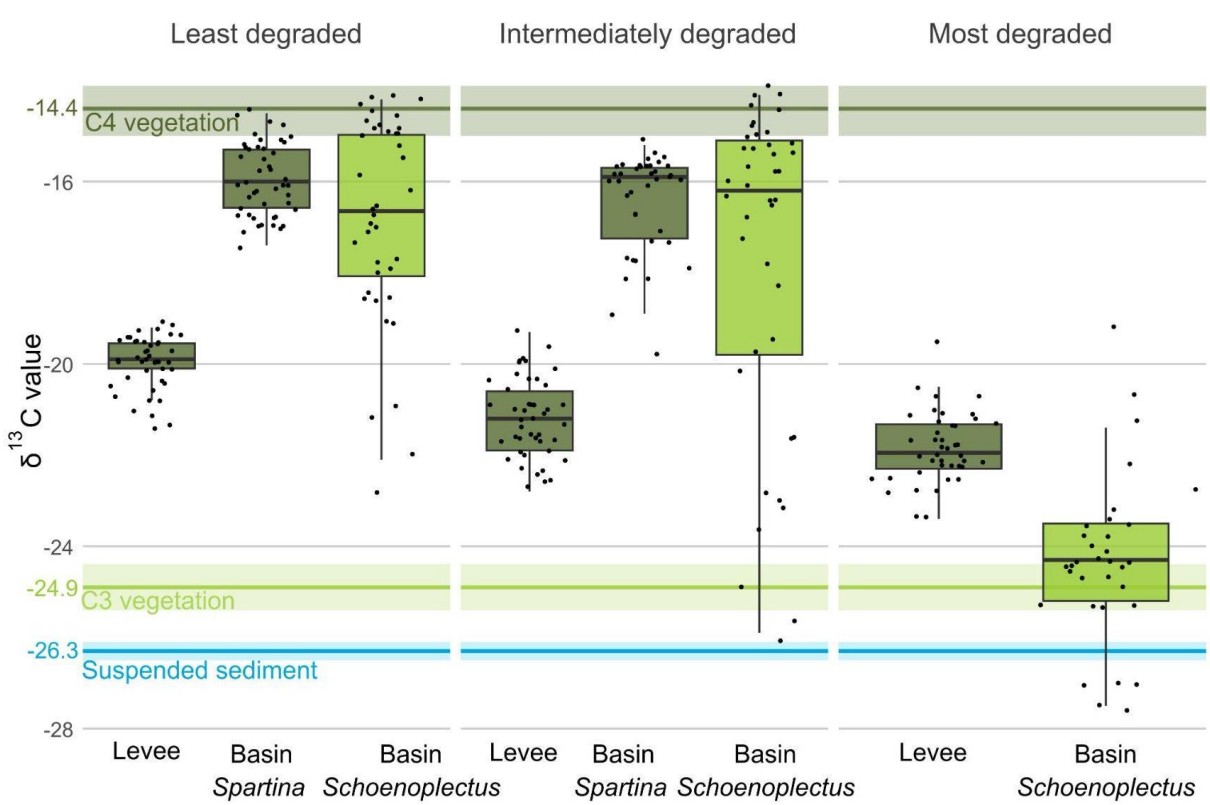

**Figure 6: $\delta^{13}$C values along the degradation and levee-basin gradient. The colours of the boxplots correspond to the photosynthetic pathway of the dominant vegetation (light green for C3, dark green for C4). The horizontal coloured lines correspond with the $\delta^{13}$C values of C3 vegetation (light**

**green), C4 vegetation (dark green) and suspended sediment (blue). The lighter-colored area around the lines correspond to the 95% confidence interval of the $\delta^{13}$C values.**

## 4 Discussion

Tidal marshes are generally known to be hotspots for organic carbon sequestration into their sediment

beds (Temmink et al., 2022; Fig. 7). However, marshes are heterogeneous landscapes where inputs of

sediment organic carbon and biogeochemistry vary across local gradients that may affect the rate at which





they accumulate organic carbon. In particular, knowledge is limited on sediment organic carbon accumulation rates (OCAR) in response to gradients in marsh degradation and levee-basin gradients. In this study, we found that marsh levees are hotspots of OCAR, accumulating organic carbon four times faster on average than in adjacent marsh basins. Based on our findings, marsh levees in a micro-tidal, organogenic marsh system appear to be among the fastest carbon accumulating environments on Earth (Fig. 7). Below, we discuss three processes that likely govern the remarkably high accumulation rates observed on marsh levees (Fig. 8): these are 1) high vegetation productivity, 2) high volumes of sediment accretion directly adjacent to tidal channels and 3) well-drained sediment beds adjacent to tidal channels, which promotes sediment compaction, creating extra accommodation space for sediment accretion.

Our results also indicate that the rate of carbon accumulation slightly increases in areas where marsh degradation is more severe. This degradation is characterised by conversion of vegetated marsh area into more unvegetated marsh area (increasing UVVR), which is considered a consequence of increasing inundation stress due to sea level rise that is not fully compensated for by marsh elevation gain. Below we discuss that the higher carbon accumulation rate is potentially related to increased inundation duration (Gonneea et al., 2019) and/or redeposition of eroded material from degraded marsh patches (Hopkinson et al., 2018).




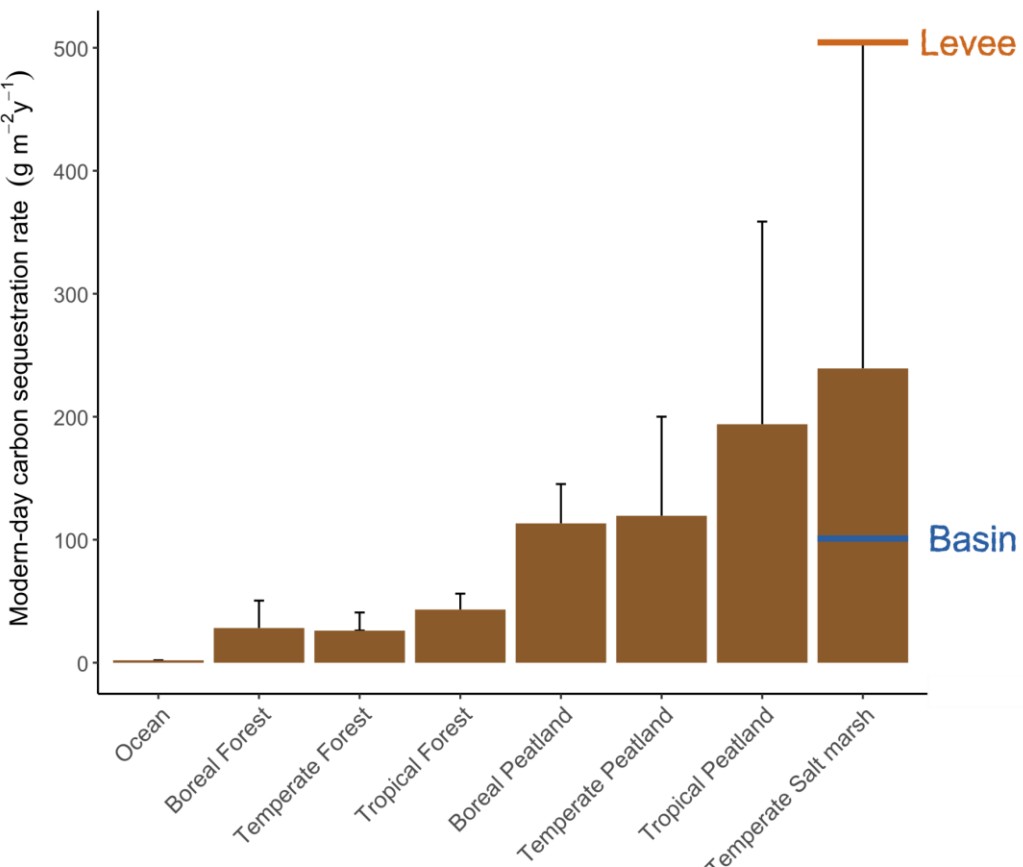

**Figure 7: Overview of the modern-day carbon sequestration rates (expressed in g C m-2 y-1) in different ecosystems (adjusted from Temmink et al., 2022), including indications of the average rates measured on our levee and basin locations. Error bars indicate the standard deviation of measurements.**



## 4.1 Higher OCAR on levees than basins

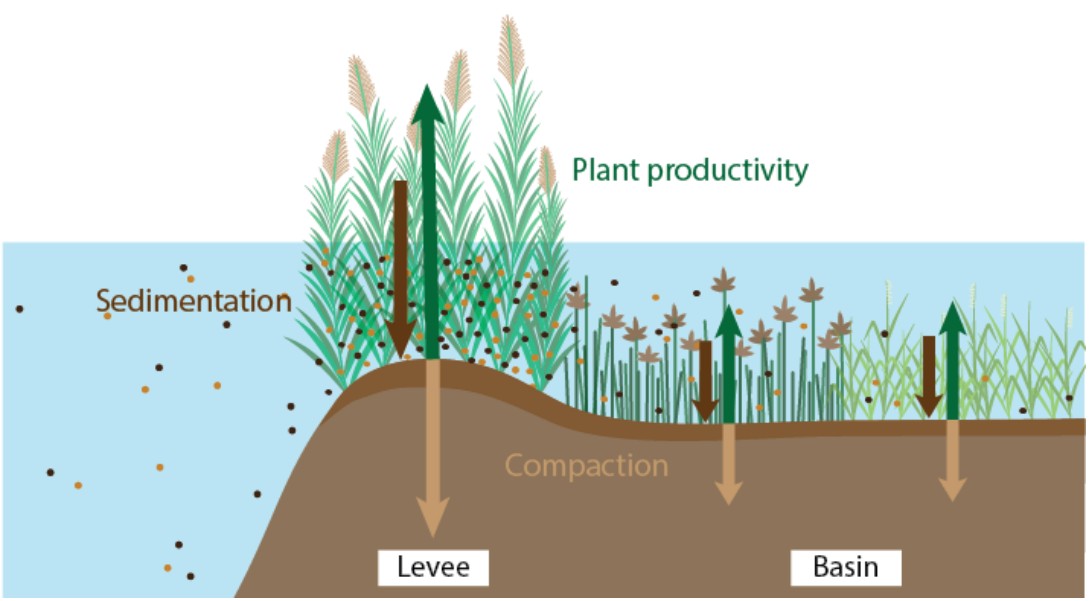

**Figure 8: Conceptual overview of processes that likely contribute to the much higher organic carbon accumulation rate on levees compared to basins. Length of the arrows indicate the gradient in sedimentation (dark brown), compaction (light brown) and vegetation productivity (green).**

### 4.1.1 Levees enhance vegetation productivity

Our results show higher aboveground vegetation biomass on the levees compared to the basins (Fig. S4). A first potential reason for this higher biomass is that soil pore water drainage during low tides is typically observed to be deeper on levees as compared to basins in tidal marshes. This is a consequence of facilitated pore water drainage towards creeks that are located next to levees, while pore water drainage from basins is hindered as they are much farther away from creeks (Armstrong et al., 1985; Balling & Resh, 1983; Mendelssohn & Seneca, 1980; Ursino et al., 2004; Van Putte et al., 2020). This deeper drainage on levees leads to better soil aeration during low tides (Mendelssohn & Seneca, 1980) and thus better conditions for vegetation growth (Callaway et al., 1997; Kirby & Gosselink, 1976). This pattern of higher vegetation biomass has been observed for multiple species, such as *Salicornia* (Balling & Resh, 1983) and *Spartina alterniflora* (Kirby & Gosselink, 1976). In our system, there is a clear species zonation between the levee, dominated by the tall *Spartina cynosuroides* and the basins, dominated by the shorter *Spartina alterniflora* or *Schoenoplectus americanus*. The higher productivity of the levees could thus also



be an intrinsic species trait of *Spartina cynosuroides* (Stalter & Lonard, 2022). A second potential reason is the higher mineral sediment content on the levees (refer to sect. 4.1.2), which has been shown to have beneficial effects on vegetation growth, such as higher availability of cations (Bricker-Urso et al., 1998; Nyman et al., 1993).

The effect of greater vegetation biomass on higher OCAR values may be twofold: more productive vegetation on levees may be expected to result in (1) more organic matter inputs into the sediment and (2) more efficient attenuation of tidal flow and related trapping of external suspended sediment delivered to the marsh during tidal inundations (Duarte et al., 2005; McLeod et al., 2011).

### 4.1.2 Levees have higher sediment accretion rates

Our results show that sediment accretion rates are higher on levees compared to basins (Fig. 2A), which is consistent with findings in other tidal marsh areas (Coleman et al., 2020; Friedrichs & Perry, 2001; Hatton et al., 1983; Reed et al., 1999; Temmerman et al., 2003), including our microtidal study area (Duran Vinent et al., 2021). In microtidal systems, such as the one investigated here, low flow velocities during high tides that inundate the marsh surface provide conditions for rapid settling of incoming suspended sediments on the marsh. Due to these low flow velocities, combined with dense vegetation on the levees, the sediment accretion rate is higher on the levees when water flows from creeks into the levee vegetation, while much less suspended sediments can reach the inner marsh basins (Reed et al., 1999). Because the suspended sediment concentration in the main tidal creek (i.e. Blackwater River) is relatively low (55 mg/l; Ganju et al., 2013)), we may hypothesize that most of the suspended sediment is deposited on the levees and the basin locations are sediment starved. This pattern in sediment deposition is confirmed by the $\delta^{13}$C value of the levee sediments (Fig. 6), where the average value (-21.0‰) indicates a mixture of different sources of carbon, from local C4 vegetation (-14.4‰) and incoming suspended sediment (-26.3‰). The basins under C4 vegetation in the least and intermediately degraded zones, however, have a $\delta^{13}$C value of (-16.2‰) that is relatively close to that of the vegetation (-14.4‰). A previous study in our study area has shown that mineral sediment deposition in basins is indeed limited and mainly occurs during storm surges (Stevenson et al., 1985). This implies that accretion in the basin locations is mostly reliant on organic matter accumulation by the local vegetation, which may explain the





much lower accretion rates in the basins versus levees. The sporadic storm tides may explain why the $\delta^{13}C$ value of the basin sediments with *Spartina* vegetation is slightly more negative than the value for the C4 vegetation.

The observed higher accretion rate on the levees results in faster burial of the carbon present in the profile, which may imply less oxygen availability to the carbon and thus lower decomposition (Rietl et al., 2021). Additionally, the suspended sediment that is deposited onto the marsh can contain substantial amounts of organic carbon. This organic carbon can originate from outside the system (e.g. from algae growth in the water or organic debris supplied with the tide) or can be the result of marsh soil material that is eroded

from elsewhere in the marsh system and redeposited (Herbert et al., 2021; Hopkinson et al., 2018). When looking at the $\delta^{13}C$ value of the suspended sediment (-26.3‰) from the river, it is relatively close to the value we found for C3 vegetation (-24.9‰), which may potentially indicate a large contribution of internally eroded marsh soil material that can be redeposited on the marsh levees.

### 4.1.3 Levees experience a higher degree of sediment compaction

Our results indicate that the bulk density is much higher on the levees compared to the basin locations (Fig. 2B). A first potential reason for that is that sediment deposition is higher on the levees than compared to the basins, resulting in a higher fraction of mineral particles on the levee (refer to sect. 4.1.2 and Coleman et al., 2020; Duran Vinent et al., 2021; Friedrichs & Perry, 2001; Hatton et al., 1983; Reed et al., 1999; Temmerman et al., 2003). This higher mineral deposition can explain the higher bulk density

as mineral sediments typically are heavier and more densely packed than organic material (Arvidsson, 1998). The results also indicate that the organic matter content of the basins is much higher than on the levees (Fig. 3), and higher sediment organic matter content is generally associated with lower sediment bulk densities (Hatton et al., 1983; Huyzentruyt et al., 2024; Nyman et al., 1993; Fig. S5). A second reason may be that the sediment on the levees is more compacted after deposition. Auto-compaction of sediments

is the process where, due to soil pore water drainage and continuous sediment deposition, water is expelled from the soil pores under the weight of the new layers, leading to contraction of soil pores and thus compaction of the sediment profile (Allen, 2000; Chen et al., 2012; Gehrels, 1999). The potential reason why levee sediments experience more compaction than basin sediments may be related to the



deeper pore water drainage during low tides on levees, because of their closer proximity to creeks, as has
been observed in many marsh studies (Armstrong et al., 1985; Balling & Resh, 1983; Mendelssohn & Seneca, 1980; Ursino et al., 2004; Van Putte et al., 2020). This higher compaction on the levees would then allow higher sediment and carbon accumulation rates by creating extra accommodation space (i.e. vertical space for sediment deposition) compared to the locations in the basin.

## 4.2 Higher OCAR in more degraded marsh zones

The results indicate that there is an increase in OCAR with increasing degree of marsh degradation in response to sea level rise, which is assessed here as an increase in the unvegetated-vegetated area ratio (UVVR). This is observed both on the levees as well as in the basin locations (Fig. 5). This result corresponds with positive relationships found between sea level rise rate and OCAR in meta-analyses based on datasets compiled from sites across continents and the globe (Herbert et al., 2021; Huyzentruyt
et al., 2024; Rogers et al., 2019; Wang et al., 2019). The main explanatory mechanism discussed in these continental- to global-scale studies is that higher sea level rise rate is associated with more marsh tidal inundation, hence higher sediment accretion rate, which drives higher OCAR. However, a major difference between our study and previous meta-data studies, is that our marsh degradation zones experience the same rate of local relative sea level rise but show different degrees of marsh degradation
in response to the sea level rise, while previous meta-data studies are based on data from different areas experiencing different rates of sea level rise. Hence an alternative explanation must be sought for the results in the Blackwater marshes.

We hypothesise that the levee and the basin in the most degraded zone may experience longer waterlogged sediment conditions, as their sediment surface elevations are lower compared to the least degraded zone
(Table 1), allowing less pore water drainage during low tides in the most degraded versus least degraded zone. This may reduce oxygenation of the sediments, thereby limiting microbial decomposition of sediment organic carbon and hence contributing to higher OCAR values. This hypothesis is also suggested by Gonneea et al. (2019) and supported by the higher levels of organic carbon content (%) that are found in the levee and basin of the most degraded versus least degraded zone (Fig. 3). Another potential
mechanism is suggested by Herbert et al. (2021), who found that marshes along the Louisiana coast with



a higher rate of marsh loss (i.e. marsh vegetation converting to ponds resulting in increase in UVVR) show a higher rate of OCAR. They hypothesise that when marsh degradation progresses, ponds form within the marsh and enlarge, which may produce eroded marsh sediment and thus organic carbon that is redistributed and redeposited in vegetated marsh zones during high tides (Hopkinson et al., 2018;

Valentine et al., 2023).  Finally, the levee of the least degraded site could be subject to the 'priming effect', where higher vegetation productivity increases the input of new carbon and oxygen into the sediment, therefore leading to higher microbial decomposition rates and thus lower overall sediment organic carbon contents (Rietl et al., 2021). This priming effect could explain why, even though vegetation is more productive in the least degraded zone than the intermediately degraded zone, the OCAR is lower (Fig.

S4). On the levees, we may expect little difference in pore water drainage along the marsh degradation gradient, as they are always located close to the creek and thus well drained, potentially explaining why there is no large difference in OCAR values between levee locations along the degradation gradient.

Besides the difference in OCAR rates we found in the levees and basins along the degradation gradient, we highlight that there is an increasing surface area of ponds (Schepers et al. 2017) within the marsh

zones with increasing degree of degradation (higher UVVR). Increasing conversion of marsh vegetation to ponds is likely to have important implications for carbon sequestration, however the processes that lead to the development and growth remain poorly understood (Redfield, 1972b; Schepers et al., 2020; van Huissteden & van de Plassche, 1998) and is one of the main remaining knowledge gaps in the carbon budget of these degrading marsh systems.

**Acknowledgements**

This paper was funded by the Research Foundation Flanders (FWO; grant number G039022N), GRG, JC, and DW acknowledge support from the U.S. Geological Survey, Ecosystems Land Change Science Program. We would like to thank Desmond Mackell and McKenna Bristow for their indispensable field assistance and Jennifer E. Connel (VIMS), Lore Fondu and Yannick Stroobandt (KULeuven) for their

assistance in the lab. Any use of trade, firm, or product names is for descriptive purposes only and does not imply endorsement by the U.S. Government.



## Data availability statement

The data used for this paper is available at:

https://zenodo.org/records/15470320?token=eyJhbGciOiJIUzUxMiJ9.eyJpZCI6IjNmZDg4YzlhLWQ4
NDktNGE1ZC04ZWU5LTFhYjNhMjNiYmZiZCIsImRhdGEiOnt9LCJyYW5kb20iOiI2NDY1NzdlMz
ViOWNhZTdmYmIyZWRkYTlhY2M1NDE2YSJ9.ms4y7BV7UBD2zGRhqbeMtx4fbTScXmLagdaV
DvU6GMYSJq6V9UdLXcYL-pHiKXCJLL19If0j-bpXVsGFRktH5A

## Author contribution statement

MH, ST conceptualised the study with the help of MK, DW, JC and GG. MH, MW, GF and DW carried
out the fieldwork and lab analysis with resources provided by GG, MK and SB. MH, MW and GF
analysed and visualised the data. MH, GF and ST prepared the manuscript with contributions from all co-
authors.

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
