# Peer review of "Carbon sequestration along a gradient of tidal marsh degradation in response to sea level rise"

_EGUsphere, 2025_

## Author Comment (AC1)

**Response to reviewer 1**

The authors present a paper describing variation in sedimentation and organic carbon accumulation between levee and basin position and along a gradient of degradation in a tidal marsh of the Chesapeake Bay ecosystem. The authors collected data from eight study sites representing levee or basin geomorphic positions and in different plant communities from the least degraded to most degraded portion of the marsh with degradation being caused by sea level rise and eventual reclamation of these wetlands by the bay. Degradation is determined by the ratio of vegetated to non-vegetated area. The authors conclude there are substantial differences in sediment deposition and carbon accumulation between levee and basin position and more modest differences along the degradation gradient. Overall, the paper is well written, clear, and easy to follow although I have some concerns I would like to see addressed.

We would like to thank the reviewer for the very thorough review of our paper. We have tried to implement your comments to the best of our abilities. In particular we (1) replaced the initial statistics with linear mixed effects models, as suggested, and (2) gave more detail on the sampling methods, such as how sites were selected, how sediment samples were cored and how replicate samples were defined. Many other smaller text changes were included in response to your feedback as detailed below.

The original feedback is denoted in black with numbered headers (1.1 for the first comment, R1.1 for the response on the first comment), followed by our response in blue. The textual changes made in the manuscript are indicated below the response (new pieces of text in blue and removed text in red).

**1.1** The authors should emphasize what is novel about the work they have conducted. Much of the paper summarizes basic physical geography of tidal wetlands. That levee positions receive more sediment is not novel – in fact, its why they are levees. Similarly, that they receive more OCAR input is simply because they receive more sediment. Plant community differences similarly are not novel – the zonation of tidal wetland communities associated with deposition and salinity are well understood. What seems most interesting is the degradation aspect of the study and how these systems change with sea-level rise. I think this theme could better come forward in the paper overall.

R1.1 This is an interesting remark. While we do agree that the geomorphic differences between levees and basins are well known and described in literature, the difference in organic carbon accumulation rate (OCAR) between these locations is very poorly described and explained in literature. In particular, we identify that much stronger compaction of the sediment bed on levees versus basins (identified from much higher sediment dry bulk density, and explained in the discussion as a likely result of better sediment pore water drainage and pore collapse on levees) creates more accommodation space for sediment and organic carbon accumulation, as such contributing to the 4-fold higher OCAR on levees versus basins. This finding is particularly novel. Further, we agree with the reviewer that the degradation aspect of the study is novel.

Therefore we aimed to emphasise both the degradation aspect and the levee-basin aspect of the study, which are both novel. We have altered the text in the introduction to highlight this more:

Line 110-113: "While the geomorphic differences between levees and basins are well known, it remains understudied to what extent the rate of organic carbon accumulation differs between both, and which processes contribute to this difference. It can be expected that these differences between basins and levees can also lead to differences in OCAR. However Moreover, there are currently no studies that have investigated the dynamics of OCAR along levee-basin gradients in marsh zones with a different degree of marsh degradation in response to sea level rise, which hampers our ability to predict the long-term stability of carbon in these systems as they progressively degrade in response to sea level rise."

**1.2** I am somewhat confused by the design of the sampling. As written in the text, the sampling appears quite limited. Table 1 and the text around 145-155 suggests 8 sampling sites and 4 soil cores per site. This would make n=32. However, the figures showing data points such as Figs. 2 and 6 suggest many, many more data points. The sampling needs to be clarified. Moreover, please describe how the sampling within a site is independent. Were samples collected along multiple transects? Minimum distance between soil cores? Overall how the soils were collected needs to be better described.

**R1.2** We have changed the explanation on the sampling design to make it clear.

Line 164-174: "The selection of study sites resulted in eight sampling locations (Table 1), two in the most degraded zone and 3 in the intermediately and least degraded zone. At each sampling location, four replicate soil cores were sampled approximately one meter apart. Three replicates were used for organic carbon analysis (see 2.3.1 and 2.3.4) and one was used for radiometric dating to determine the sediment accretion rate (see 2.3.2). As a result, of the total of 32 cores, 8 were used for radiometric dating and the remaining 24 for organic carbon analysis. Every core was between 25 and 50 cm long and was sliced in increments of about 1 cm. For the organic carbon analysis every other depth interval was used, leading to 12 to 25 data points for each core and a total number of 329 data points. This resulted in a total of 8 sampling locations (Table 1). At each sampling location four replicate cores were taken, of which one was used for radiometric dating (refer to section 2.3.2) and three were used for bulk density and organic carbon analysis."

In the initial figure 2 and figure 6 we show all measured depth intervals of every sampled core, which indeed results in more datapoints than sampled cores. Based on this comment and that of reviewer 2, we have changed our figures to show both the average value and standard deviation of each core:

Figure 1: Dry bulk density (left) and sediment accretion rates (right) determined with radiometric dating along the levee basin gradient. The coloured points indicate the average value for all depth measurements of each core and the error bars show the standard deviation for all depth measurements of each core. The different shapes indicate data from the least degraded (square), intermediately degraded (triangle) and most degraded (circle) zone. The letters above indicate the significance of the differences between levee, basin *Spartina* and basin *Schoenoplectus*, where observations with the same letters are not significantly different from each other (derived from ANOVA for the sediment accretion and from linear mixed models for bulk density).

and

Figure 6:  $\delta 13$ C values along the degradation and levee-basin gradient. The coloured points indicate the average value for all depth measurements of each core and the error bars show the standard deviation for all depth measurements of each core. The colours of the boxplots points correspond to the photosynthetic pathway of the dominant vegetation (light green for C3, dark green for C4). The horizontal coloured lines correspond with the  $\delta 13$ C values of C3 vegetation (light green), C4 vegetation (dark

green) and suspended sediment (blue). The lighter-coloured area around the lines correspond to the 95% confidence interval of the  $\delta 13C$  values.

- **1.3** Similarly, I am concerned about the sampling of the gradient in degradation. The gradient is described/quantified as the ratio of vegetated to non-vegetated surface and while this ratio is reported in table 1, the sampling as I understand it is somewhat misleading since only the vegetated portions of the marsh were sampled, regardless of gradient position. Clearly the vegetated and non-vegetated portions of the marsh would experience differences in OCAR input so the decision to only sample vegetated i.e., least degraded regardless of the degradation gradient needs to be justified and the implication of this choice clearly described.
- **R1.3** We did in fact try to sample the unvegetated shallow ponds within the marsh as well, but this turned out to be unsuccessful. The sediment at the bottom of the unvegetated ponds was very loose, unconsolidated, fluid mud, making it impossible to sample solid sediment cores using the same method as for the cores sampled from the vegetated marsh portions (using coring tubes), where the sediment was consolidated and bound together by roots. However, the remaining vegetated marsh portions are also very different along this gradient of UVVR. In the most degraded zone the vegetated marsh portions have sediment beds that are much less strong (clearly noticeable as our feet could sink up to several dm into the sediment bed during the field work) as compared to the least degraded zone (where we did not sink much into the sediment bed). We added an explanation in the methods section to explain why only the vegetated parts were sampled.

Line 154-15: "Degraded zones consist of a mosaic of vegetated marsh portions and large pools of open water, the latter having sediment beds consisting of fluid mud where sampling fixed sediment volumes was not feasible. Therefore, we sampled only vegetated marsh sediment beds in each zone. Within each zone, samples were collected on levee and on vegetated basin locations."

- **1.4** The description of the statistical analysis is too limited for the statistical procedure to be evaluated. Please expand the analysis section to indicate if fixed or mixed models were used and any random effects, any data transformations, selection of post-hoc tests (the results of which are show in the figures).
- **R1.4** You are indeed correct. We changed our statistical analysis to linear mixed models, where core is used as a random effect combined with a Tukey post-hoc test to see the differences between all the sites and locations. We have changed the section on the statistical analysis as follows:

Line 242-250: "For sediment accretion rates and bulk density, the difference between levee and basin locations with *Schoenoplectus* and *Spartina* was investigated using ANOVA in R version 4.4.1 (R Core Team, 2022). For the organic carbon content, density and accumulation rate, the separate effects of degradation zone and location (basin or levee) were investigated using linear mixed effects models, including core and depth as random factors, using the lme4 package (Bates et al., 2015). Besides the simple

effect of location and degradation zone, we ran an additional model with their interaction effect. To see which locations and zones differed from each other, a Tukey post-hoc test was done using the emmeans package in R (Lenth, 2025). Bulk density was analysed in a similar way, but only looking at the difference between levee and basin locations. , the average value of each core was used to minimize the effect of the depth profiles. The difference between levee and basin locations and zones with a different degree of marsh degradation were also investigated using ANOVA."

**1.5** I find several inconsistencies in the arguments surrounding the differences between the levee and basin communities. Line 380 suggests that high accretion rates in levees may be due to rapid burial of organic matter and low O2 availability leading to lower decomposition. However, on line 339, there is the suggestion that deep-soil pore water drainage on levees promotes oxygenation and more rapid plant growth. While perhaps these can both be true depending on the precise depth of anoxia, its reads as inconsistent. Similarly, on line 394, the packing of high-density mineral matter on the levees is used as a justification for the greater bulk density of levee soils would further argue against rapid drainage and oxygenation.

**R1.5** Thank you for bringing this to our attention. We understand that the processes we're describing can be seen as inconsistent, but it is indeed the depth of the anoxia that plays a role. Since the accretion rates are high on the marsh levees, we argue that this will result in faster burial of the present carbon to layers below the oxic zone, even though this oxic zone is deeper on the levees compared to the basins. We have highlighted this in the text as follows:

Line 432-437: "Even though it may be expected that sediment pore water drainage is deeper in levees (Armstrong et al., 1985; Balling & Resh, 1983; Mendelssohn & Seneca, 1980; Ursino et al., 2004; Van Putte et al., 2020), the observed higher accretion rate on the levees results in faster burial of the carbon present in the profile, so that it may faster reach layers below the sediment drainage level, where oxygen is less available. This could imply lower rates of decomposition and thus better preservation of the present carbon which may imply less oxygen availability to the carbon and thus lower decomposition (Rietl et al., 2021)."

1.6 I have concerns with the interpretation of the 13C data as presented here. The sediment varies considerably in 13C suggesting different sources of OCAR input as the authors indicate. However, the endpoints of the carbon is somewhat ambiguous. The argument is made that 13C can determine the difference between autochthonous C and allochthonous C. However, autochthonous C can come from two sources – the C4 grass *Spartina* and C3 rush *Schoenoplectus* while allochthonous C is assumed to be C3 (presumably phytoplankton and other algae). Therefore, while seeing a highly C4 signature in sediment is good indication of local C in a *Spartina* zone the opposite is not necessarily true since the deposition could be from OCAR input from outside the wetland as well as OCAR input from remobilized sediment with a local source. Please address this concern in interpreting these data. Figure 6 I think shows the community shift happening with the *Schoenoplectus* OCAR being mostly C4-derived in the least degraded and intermediate sites and mostly C3 derived in the most degraded. Since this is a C3 plant,

the data suggest a recent conversion (and the large error bars support this) in the least and intermediate sites but a long-term history of the C3 rush in the most degraded. Combined with the assertion that basins are sediment starved, the data argue for local carbon inputs dominating the basin system.

**R1.6** This is a very valid suggestion. We have incorporated it in the manuscript.**

Line 419-426: "...This pattern in sediment deposition is confirmed by the  $\delta^{13}$ C value of the levee sediments (Fig. 6), where the average value (-21.0‰) indicates a mixture of different sources of carbon, from local C4 vegetation (-14.4‰) and incoming suspended sediment (-26.3‰). The basins under C4 vegetation in the least and intermediately degraded zones, however, have a  $\delta^{13}$ C value of (-16.2‰) that is relatively close to that of the vegetation (-14.4‰). Hence for these basin locations we can conclude the sediment organic carbon mainly originates from autochthonous input by the local C4 vegetation. For the basin under C3 vegetation, i.e. in the most degraded zone, we cannot be sure whether the soil organic carbon is mainly from autochthonous origin, since the  $\delta^{13}$ C signature of the local C3 vegetation is close to that of allochthonous suspended sediment. ..."

As for figure 6 we agree that a vegetation shift may have happened from Spartina dominated to Schoenoplectus dominated, however this discussion is speculative and does not provide an added value to the focus of this paper on carbon accumulation (since we observed no significant differences in sediment properties nor organic carbon content/density/accumulation rate between basin sites with Spartina or Schoenoplectus), so we decided to leave it out of this paper.

---

## Author Comment (AC2)

**Response to reviewer 2**

**General Comments**

Thank you for the opportunity to review the manuscript titled "Carbon sequestration along a gradient of tidal marsh degradation in response to sea level rise" by Mona Huyzentruyt and colleagues. This paper reports on the differences in organic carbon accumulation rates among levees and basins along a marsh degradation gradient within a microtidal wetland. The results of the paper indicate that carbon accumulation rates are much greater on the levees than within the basins. Some differences in organic carbon accumulation rates were detected among the different marsh degradation zones which may suggest accumulation rates tend to be greater in more degraded zones.

We would like to thank the reviewer for the very thorough review of our paper. We have tried to implement your comments to the best of our abilities. In particular we (1) replaced the initial statistics with linear mixed effects models, as suggested, and (2) gave more detail on the sampling methods, such as how sites were selected, how sediment samples were cored and how replicate samples were defined. Many other smaller text changes were included in response to your feedback as detailed below.

The original feedback is denoted in black with numbered headers (2.1 for the first comment, R2.1 for the response on the first comment), followed by our response in blue. The textual changes made in the manuscript are indicated below the response in separate boxes (new pieces of text in blue and removed text in red).

The manuscript represents a substantial contribution to scientific progress, and important new data, in the Biogeosciences. The paper is well organized and written without errors but could be improved by editing to reduce text, and re-arranging where some information is introduced within the text.

2.1 There are a few issues within the statistical approach that can be addressed and will likely not have a large effect on the main results reported by the paper. Specifically, there are several analyses (dry bulk density and C-13) where it appears all of the subsamples from all cores were included as individual observations and run in an ANOVA. If this is the case then this method would artificially inflate the sample size and therefore artificially decrease the error term, as each sample is being treated as independent even though multiple samples came from the same core. To deal with this, the authors could consider a repeated measures ANOVA or linear mixed modeling approach where "depth" can be nested within "core". It could be that the authors want to consider taking this approach in other analyses as well to potentially gain a little more insight from the data they have, as it could help take advantage of all the depth data rather than just averaging it all to one value. But the other analyses are ok as is if the authors don't want to make that change.

**R2.1** This is a very valid remark. We have altered the statistical analysis to linear mixed effects models, using both depth and core as random factors. We tested the effect of degradation zone and location (levee or basin) separately as well as their interaction.

Line 258-268: "For sediment accretion rates and bulk density, the difference between levee and basin locations with *Schoenoplectus* and *Spartina* was investigated using ANOVA in R version 4.4.1 (R Core Team, 2022). For the organic carbon content, density and accumulation rate, the separate effects of degradation zone and location (basin or levee) were investigated using linear mixed effects models, including core and depth as random factors, using the lme4 package (Bates et al., 2015). Besides the simple effect of location and degradation zone, we ran an additional model with their interaction effect. To see which locations and zones differed from each other, a Tukey post-hoc test was done using the emmeans package in R (Lenth, 2025). Bulk density was analysed in a similar way, but only looking at the difference between levee and basin locations. The difference between levee and basin locations and zones with a different degree of marsh degradation were also investigated using ANOVA."

As for the  $\delta^{13}$ C values, we did not perform any statistical testing on this data, we only show it to compare the soil carbon signatures to those of the incoming sediment and local vegetation. This gives us insight in the sources of carbon within the system. We have clarified this in the statistical analysis paragraph.

Line 268-270: "No statistical testing was done on the  $\delta$ 13C values, but they were used to estimate the origin of the sediment organic carbon (autochthonous versus allochthonous), by comparing  $\delta$ 13C values between the soil organic carbon values, the local vegetation values and the external suspended sediment values."

- **2.2** The authors also make the claim that the degraded marsh zones are experiencing the same rate of relative sea level rise as the other zones, but this should be better discussed and documented as they also make statements that suggest the degraded zone may be decreasing in elevation (more ponds, etc) which would mean that the rate of relative sea level rise may be greater in the degraded marsh zones.
- **R2.2** Following the established literature on tidal marsh responses to sea level rise, we refer to 'relative sea level rise' as the **regional** relative sea level rise, here for the broader Chesapeake Bay region, which is resulting from the combination of geocentric (eustatic) sea level rise and land subsidence (dominated by glacial isostatic adjustment for the Chesapeake Bay region). With relative sea level rise we do not mean the very local changes in tidal inundation experienced within local marsh zones (such as the most, intermediate and least degraded marsh zones). The latter is, apart from regional relative sea level change, also affected by very local marsh surface elevation change (resulting from local sediment accretion, local erosion, local shallow sediment compaction). This is clarified now in the text:

Line 475-479: "However, a major difference between our study and previous meta-data studies, is that our marsh degradation zones experience the same rate of regional relative sea level rise (i.e. for the Chesapeake bay region) but show different degrees of local marsh degradation in response to the regional relative sea level rise, while

previous meta-data studies are based on data from geographically distant different areas experiencing different rates of relative sea level rise. ..."

**Additionally, we added some explanation in the materials and methods:**

Line 135-143: "The part of the Chesapeake Bay closest to the Blackwater marshes experiences a regional relative sea level rise rate of 4.06 mm y-1 (measured between 1943 and 2024; NOAA station Cambridge, MD, https://tidesandcurrents.noaa.gov/sltrends/, accessed on 6/30/2025), which is higher than the average historical sediment accretion rate of 3.9 mm y-1 measured in the Blackwater marshes (Ganju et al., 2013). This accretion deficit This average accretion deficit has led to severe marsh degradation. The spatial gradient in tidal range and marsh elevation (Table 1) along the river results in different tidal inundation regimes at the different marsh locations. This has led to a spatial gradient in marsh degradation, with undegraded marshes close to the Fishing Bay and increasing historical conversion of marsh to ponds moving upstream along the Blackwater River (Schepers et al., 2017)."

**Specific Comments:**

**Graphical abstract**

- **2.3** In the graphical abstract – suggest indicating the direction of the coast is given that the setting is describing tidal marsh ecosystems.

**R2.3** We added arrows indicating the direction of the mainland areas and of the Chesapeake Bay.

**Abstract**

**2.4** In line 24 the authors state: "Additionally, OCAR was observed to increase with increasing degree of marsh degradation in response to sea level rise" but really there is just one difference in the basin rates ('most degraded' is higher) and one difference in the levee rates ('least degraded' is lower).

**R2.4** It is indeed correct that only one levee and one basin rate are different, however for the readability and simplicity of the abstract we have decided to keep it like this. But we have added more nuance in the discussion.

Line 468-473: "This is observed both on two of the levees as well as in one of the basin locations (Fig. 5). It is however important to note that only three points along the degradation gradient were measured, so general conclusions should be made with caution. However, This result does corresponds with positive relationships found between sea level rise rate and OCAR in meta-analyses based on datasets compiled from sites across continents and the globe (Herbert et al., 2021; Huyzentruyt et al., 2024; Rogers et al., 2019; Wang et al., 2019).

**Introduction**

- **2.5** Does 'marsh degradation' specifically mean less vegetation? If so, make sure this is clearly defined in the Introduction.

**R2.5** The most obvious sign of marsh degradation is die off of vegetation resulting in bare soil or even ponds (as specified in line 61-66). However, there are also changes in the stability of sediment. In the most degraded zone the vegetated marsh portions have sediment beds that are much less strong (clearly noticeable as our feet could sink up to several dm into the sediment bed during the field work) as compared to the least degraded zone (where we did not sink much into the sediment bed). Since this is something we observed, and not reported so far in literature, we can not write about it in the introduction, but we emphasise it in the materials and methods.

Line 154-157: "Degraded zones consist of a mosaic of vegetated marsh portions and large pools of open water, the latter having sediment beds consisting of fluid mud where sampling fixed sediment volumes was not feasible. Therefore, we sampled only vegetated marsh sediment beds in each zone. Within each zone, samples were collected on levee and on vegetated basin locations."

**2.6** How widespread is the phenomenon of decreasing vegetation in tidal marshes globally?

**R2.6** Vegetation loss in tidal marshes can be caused by many processes other than sea level rise. However, the systems that we list in our introduction (line 58-60) are all systems that are experiencing losses in vegetation or surface elevation due to the inability to keep up with sea level rise, which is a globally widespread phenomenon experienced specifically in microtidal marshes (specified line 55-58). Including other sources of marsh loss, it is predicted that global wetland loss will be between 0-30% of the current area by 2100 (Schuerch et al., 2018). We added this global estimate to the introduction:

Line 60-62: "A global scale study has estimated that coastal wetland (mangrove and marsh) loss will range between 0 and 30% by 2100 (Schuerch et al., 2018)."

- **2.7** How were the degradation zones (least degraded, intermediately degraded, most degraded) determined? Now I see this is answered in the supplement, but a brief explanation should be included in the paper text (the material in the supplement can remain the same).
  - **R2.7** Thank you for bringing this to our attention. We do bring on the concept of the Unvegetated-vegetated ratio already in the introduction (line 65-67), this ratio was used to determine our degradation zones. But we have added additional clarification in the fieldwork setup paragraph:

Line 149-153: Three marsh zones were selected along the marsh degradation gradient, based on an increasing unvegetated-vegetation ratio (UVVR, Ganju et al., 2017). These sites will further be referred to as (1) least degraded (UVVR=0), (2) intermediately degraded (UVVR=0.016) and (3) most degraded (UVVR=0.143) (Fig. 1, Fig. S1, Table 1). This gradient is characterized by an increasing unvegetated-vegetated ratio (UVVR; Ganju et al., 2017) moving from the least to most degraded zone (Fig S1, Table 1).

- **2.8** Introduce the difference between C3 and C4 vegetation and why it matters in this context in the introduction (as they were investigated separately in this work).

**R2.8** We have decided to integrate the relevance of sampling both C3 and C4 vegetation in the method section rather than the introduction. We believe it would disrupt the storyline of the introduction. We indicated the changes made to the method section in our reply to comment 2.10.

**Methods**

- **2.9** It seems that four cores were taken within each zone, but at the same site. Why weren't cores taken from multiple sites throughout the zone... that seems like it would better represent the carbon dynamics of that zone.

**R2.9** Indeed all 4 cores were taken at the same location, with at least 1 meter distance between them. This was chosen to obtain one core for radiometric dating and three replicate cores for bulk density and organic carbon analysis, as explained in the methods section. We did it this way to limit the number of field sites and thus time needed in the field to go to each site. Given the time we had available for the field work, this was the maximum number of sampling locations and cores we could sample. We have specified the sampling design more in the materials and methods.

Line 163-173: "The selection of study sites resulted in eight sampling locations (Table 1), two in the most degraded zone and 3 in the intermediately and least degraded zone. At each sampling location, four replicate soil cores were sampled approximately one meter apart. Three replicates were used for organic carbon analysis (see 2.3.1 and 2.3.4) and one was used for radiometric dating to determine the sediment accretion rate (see 2.3.2). As a result, of the total of 32 cores, 8 were used for radiometric dating and the remaining 24 for organic carbon analysis. Every core was between 25 and 50

cm long and was sliced in increments of about 1 cm. For the organic carbon analysis every other depth interval was used, leading to 12 to 25 data points for each core and a total number of 329 data points. This resulted in a total of 8 sampling locations (Table 1). At each sampling location four replicate cores were taken, of which one was used for radiometric dating (refer to section 2.3.2) and three were used for bulk density and organic carbon analysis."

**2.10** Explain why two vegetation types were sampled. Perhaps the C3 vs C4 difference should be introduced in the Introduction if the authors think it is important.

**R2.10** We do believe that this is an important point. The first reason that two vegetation species are sampled along the gradient is because there is a clear mosaic of patches dominated by one or the other species (which is specified in the methods Line 146) along a big part of the gradient and we wanted to see whether there was a difference between these species in terms of carbon dynamics. Second, we wanted to be able to look at the source of the carbon and not only the accumulation rate. Since there is a larger difference in  $\delta^{13}$ C value between C4 vegetation and incoming sediment, it makes the distinction between sources easier than with C3 vegetation. We have specified this in the method section:

Line 157-161: "Because the basins of the least degraded and intermediately degraded zone contained distinct patches of two vegetation types, samples were taken within these zones at two basin locations, i.e. in each of the two vegetation types (one dominated by *S. americanus*, a C3 species and the other by a mixture of *S. alterniflora* and *S. patens*, C4 species), but only one levee location was sampled (dominated by *S. cynosuroides*, a C4 species)."

**and**

Line 240-245: "The  $\delta^{13}$ C values were also measured for above-ground vegetation, by analysing finely ground vegetation samples, and for suspended sediment samples. The  $\delta^{13}$ C signature of C3 and C4 vegetation is very different (Bouillon & Boschker, 2006; Farquhar et al., 1989), with C4 vegetation typically having a signature around -14‰, and C3 around -26‰ (Bouillon & Boschker, 2006). Since incoming sediment often has a  $\delta^{13}$ C signature similar to C3 vegetation, it is more straightforward to distinguish between vegetation and externally derived carbon within C4 vegetation. For the analysis of the suspended sediment samples, the filters were cut into four equal parts and one part was used for the  $\delta^{13}$ C analysis."

- **2.11** Why were water samples taken at just one location and how is this information used? Was the site inundated, and this was the water present above the soil surface?

**R2.11** The water sample was collected in the tidal creek. Since we are working in a microtidal system, tidal inundation is limited, and all sediment deposition comes from the tidal creek. The water sample was analysed for carbon isotopes, identify the signature of the suspended sediment and compare this to the sediment carbon isotope signatures. We have clarified both the sampling and the use of the samples in the text.

Line 220-224: "Water samples were collected to analyse the suspended sediments for their  $\delta^{13}$ C value (see 2.3.4), to be able to evaluate whether the sediment organic carbon was mainly plant derived or coming from the tidal deposition of suspended sediments. As the delivered sediment comes from the tidal channel, \text{\text{W}} water samples were collected at one location \text{\text{along}} from the tidal channel (Blackwater River). \text{\text{and}} After collection, the samples were stored in the fridge until further analysis."}

**Statistical Analyses**

- 2.12 The description of the stats leaves some questions. Were levee and basin sites from all of the degradation zones all analyzed against each other, or were levee and basin nested within zone?
  - **R2.12** In our initial analysis, we analysed them all against each other. However, in the new analysis (see below), we have changed the statistics to linear mixed models and included the effect of site and location separately as well as the interaction between both. This way we could see the overall difference in location and site as well as the specific differences between each of the combinations. (see also R2.1 and R2.13).
- 2.13 Figure 2. It is not statistically appropriate to pool values from different depths within one core as they are not independent. However, the authors could address this by adopting a linear mixed effects modeling framework that nests "depth" within "core". Such an approach may actually be useful for some of the subsequent analyses as well because it may help the authors determine more about the differences in carbon density, for example, in the basin sites along the degradation gradient that currently are all the same using the ANOVA analyses, but the differences may be parsed out if all of the measurements from each core were included. It may be overkill as the authors do not specifically have 'depth' questions but as the analyses are set up right now a lot of information (and work!!) is being 'tossed out' as the cores are averaged to just one value.

**R2.13** You are indeed correct. We have repeated the statistical analysis with linear mixed models instead of just ANOVA testing. It has been changed in the method section (as in response to R2.1).

And the figure captions have also been changed.

- **2.14** Lines 266-270, as the authors point out, there is no statistical difference in the mean carbon density among the basins of different degradation zones, so that cannot be reported as a finding.

**R2.14** You are indeed correct. We have rephased the section:

Line 305-311: "For organic carbon densities (g cm-3; OCD; Fig. 4) the values were significantly higher (p < 0.05) on the levees compared to the basins, for all zones along the marsh degradation gradient. There was however no significant difference between the values in the basins or levees of the different zones (Fig. 4). The values in the basins also increased along the degradation gradient, with the highest basin values found in the most degraded zones (0.030 g cm-3), followed by the intermediately degraded basin with Schoenoplectus (0.029 g cm-3) and Spartina (0.028 g cm-3). The lowest values were found in the least degraded basin with Schoenoplectus (0.025 g cm-3) and with Spartina (0.024 g cm-3)."

**2.15** Introduce the C3 vs C4 difference among the two vegetation types earlier as the reasoning for separating them and explain why this is important in introduction.

**R2.15 See response to comment 2.10.**

2.16 Why are the individual depth measurements used for the C-13 plot and not for the others, and how did the authors avoid pseudo-replication? (Does the statistical model include depth nested within core to avoid inflating the sample size and artificially shrinking the error term?) Remember that multiple depths are essentially 'repeated measures' within a core.

**R2.16** We did not do any statistical testing on the  $\delta^{13}$ C values, as they are not meant to compare between the different sites, but to compare the soil carbon  $\delta^{13}$ C values with the  $\delta^{13}$ C values of the incoming suspended sediment and the local vegetation. We have added a clarification in the method section and added the remark to the caption of figure 6.

Line 252-254: "No statistical testing was done on the  $\delta$ 13C values, but they were used to estimate the origin of the sediment organic carbon (autochthonous versus allochthonous), by comparing  $\delta$ 13C values between the soil organic carbon values, the local vegetation values and the external suspended sediment values."

and

Figure 6:  $\delta 13C$  values along the degradation and levee-basin gradient. The coloured points indicate the average value for all depth measurements of each core and the error

bars show the standard deviation for all depth measurements of each core. The colours of the points boxplots—correspond to the photosynthetic pathway of the dominant vegetation (light green for C3, dark green for C4). The horizontal coloured lines correspond with the  $\delta$ 13C values of C3 vegetation (light green), C4 vegetation (dark green) and suspended sediment (blue). The lighter-coloured area around the lines correspond to the 95% confidence interval of the  $\delta$ 13C values.

**Discussion**

- **2.17** Figure 7 is unnecessary.

**R2.17** Thank you for this remark. We do believe that this figure adds value by placing our results in a larger global context, rather than only in a tidal marsh context. Therefore, we would like to keep this figure in the manuscript.

2.18 Line 312 states that the study examines: "accumulation rates (OCAR) in response to gradients in marsh degradation and levee-basin gradients." It seems there is an important distinction between examining accumulation rates across gradients of degradation, and "in response" to degradation. It seems the authors are doing the former and therefore should use that language here, i.e. change to "accumulation rates across gradients in marsh degradation."

**R2.18** Thank you for noticing our error, we have changed the sentence accordingly:

Line 358-360: "In particular, knowledge is limited on sediment organic carbon accumulation rates (OCAR) along in response to gradients in marsh degradation and levee-basin gradients. In this study, we found that marsh levees are hotspots of OCAR,..."

2.19 Line 313, what is the relative area of levees to basins in this wetland, and in most tidal wetlands? This will help provide context on the relative importance of these 'hotspots'

**R2.19** In our study area, the levees are between 10 and 20 m wide depending on where along the Blackwater river you are. We are currently working on a spatial study to estimate how the total carbon budget of a system is influenced by taking into account or disregarding the effect of levees. This is also mentioned in the introduction (Line 97-98) and the materials and methods (line 129). We have also added it in our discussion.

Line 361-363: "In this study, we found that marsh levees are hotspots of OCAR, accumulating organic carbon four times faster on average than in adjacent marsh basins. Even though their area is limited (in this case a band of 10-20m width along the river), we believe that taking the difference in carbon accumulation rate between levees and basins into account can make a big difference for system-scale carbon estimates."

- **2.20** Line 315: when the authors state that levees are "among the fastest carbon accumulating environments on Earth" they are talking specifically about soil organic carbon accumulation, right?

**R2.20** We do indeed mean soil organic carbon accumulation and have clarified it as such in the text.

Line 364-365: "Based on our findings, marsh levees in a micro-tidal, organogenic marsh system appear to be among the fastest **soil** carbon accumulating environments on Earth (Fig. 7)."

2.21 Figure 8. Clear and relatively easy to understand but at first glance the relative size of the arrows among the levee and the two vegetation communities appears to be the same. It is difficult to determine the 'point' of the conceptual figure – are there differences in the relative strength of these processes among the different locations? It seems there must be if the accumulation rates are so different, but it is difficult to see this from the figure.

**R2.21** We have changed the thickness of the arrows (instead of only the height) to highlight the difference in relative strength of the processes.

**2.22** Line 410: Be specific that one basin rate differs from the other two and one levee rate differs from the other two. Especially given that these rates were only sampled in one location per zone (via three cores), it seems to be overstating the results a bit to claim that there is an increase in OCAR with increasing marsh degradation.

**R2.22** You are correct that a bit more nuance is warranted. We have changed the sentence as follows:

Line 468-473: "This is observed both on two of the levees as well as in one of the basin locations (Fig. 5). It is however important to note that only three points along the degradation gradient were measured, so general conclusions should be made with caution. However, This result does corresponds with positive relationships found between sea level rise rate and OCAR in meta-analyses based on datasets compiled from sites across continents and the globe (Herbert et al., 2021; Huyzentruyt et al., 2024; Rogers et al., 2019; Wang et al., 2019).

2.23 Line 419: This is interesting - do the authors know that the degraded marsh experiences the same rate of relative sea level rise? It seems that it could be slightly different given that vegetation has been lost so perhaps rates of accretion are lower? If the degraded area is experiencing any subsidence, or even just lower rates of accretion, then it would be experiencing a faster rate of relative sea level rise.

**R2.23** Following the established literature on tidal marsh responses to sea level rise, we refer to 'relative sea level rise' as the **regional** relative sea level rise, here for the broader Chesapeake Bay region, which is resulting from the combination of geocentric (eustatic) sea level rise and land subsidence (dominated by glacial isostatic adjustment for the Chesapeake Bay region). With relative sea level rise we do not mean the very local changes in tidal inundation experienced within local marsh zones (such as the most, intermediate and least degraded marsh zones). The latter is, apart from regional relative sea level change, also affected by very local marsh surface elevation change (resulting from local sediment accretion, local erosion, local shallow sediment compaction). This is clarified now in the text:

Line 135-143: "The part of the Chesapeake Bay closest to the Blackwater marshes experiences a regional relative sea level rise rate of 4.06 mm y-1 (measured between 1943 and 2024: NOAA station Cambridge, MD, 8571892, https://tidesandcurrents.noaa.gov/sltrends/, accessed on 6/30/2025), which is higher than the average historical sediment accretion rate of 3.9 mm y-1 measured in the Blackwater marshes (Ganju et al., 2013). This accretion deficit This average accretion deficit has led to severe marsh degradation. The spatial gradient in tidal range and marsh elevation (Table 1) along the river results in different tidal inundation regimes at the different marsh locations. This has led to a spatial gradient in marsh degradation, with undegraded marshes close to the Fishing Bay and increasing historical conversion of marsh to ponds moving upstream along the Blackwater River (Schepers et al., 2017)."

**and**

Line 475-479: "However, a major difference between our study and previous meta-data studies, is that our marsh degradation zones experience the same rate of regional relative sea level rise (i.e. for the Chesapeake bay region) but show different degrees of local marsh degradation in response to the regional relative sea level rise, while previous meta-data studies are based on data from geographically distant different areas experiencing different rates of relative sea level rise. ..."

2.24 Lines 430-435: all of the processes described between Line 430 and 435 indicate that
degraded marshes do experience sediment loss which would then make them vulnerable
to higher rates of relative sea level rise.

**R2.25** No this is a misinterpretation. We refer to our reply R2.2 and R2.23 above. To clarify our point in the manuscript, we changed 'relative sea level rise' into 'regional relative sea level rise'.

**Technical Corrections:**

- **2.25** Section 4.1.1. Minor writing suggestion - three sentences in a row start with "This", consider rephrasing to reduce redundancy. The section could likely also be condensed.

**R2.25** Thank you for noticing. We have changed the paragraph as follows:**

Line 393-397: "This is a consequence of facilitated pore water drainage towards creeks that are located next to levees, while pore water drainage from basins is hindered as they are much farther away from creeks (Armstrong et al., 1985; Balling & Resh, 1983; Mendelssohn & Seneca, 1980; Ursino et al., 2004; Van Putte et al., 2020). This The deeper drainage on levees leads to better soil aeration during low tides (Mendelssohn & Seneca, 1980) and thus better conditions for vegetation growth (Callaway et al., 1997; Kirby & Gosselink, 1976). This pattern of higher vegetation biomass has been observed Other studies have found a similar pattern for multiple species, such as *Salicornia* (Balling & Resh, 1983) and *Spartina alterniflora* (Kirby & Gosselink, 1976)."

- 2.26 Line 355: remove "be expected to"

R2.26 We removed it.

- 2.27 Line 368: remove one parenthesis after "Ganju et al., 2013"

**R2.27** Thank you for noticing, we removed the extra parenthesis.

---

## Author Response (AR1)

**Response to reviewer 1**

The authors present a paper describing variation in sedimentation and organic carbon accumulation between levee and basin position and along a gradient of degradation in a tidal marsh of the Chesapeake Bay ecosystem. The authors collected data from eight study sites representing levee or basin geomorphic positions and in different plant communities from the least degraded to most degraded portion of the marsh with degradation being caused by sea level rise and eventual reclamation of these wetlands by the bay. Degradation is determined by the ratio of vegetated to non-vegetated area. The authors conclude there are substantial differences in sediment deposition and carbon accumulation between levee and basin position and more modest differences along the degradation gradient. Overall, the paper is well written, clear, and easy to follow although I have some concerns I would like to see addressed.

*We would like to thank the reviewer for the very thorough review of our paper. We have tried to implement your comments to the best of our abilities. In particular we (1) replaced the initial statistics with linear mixed effects models, as suggested, and (2) gave more detail on the sampling methods, such as how sites were selected, how sediment samples were cored and how replicate samples were defined. Many other smaller text changes were included in response to your feedback as detailed below.*

*The original feedback is denoted in black with numbered headers (1.1 for the first comment, R1.1 for the response on the first comment), followed by our response in blue. The textual changes made in the manuscript are indicated below the response (new pieces of text in blue and removed text in* red*).*

**1.1** The authors should emphasize what is novel about the work they have conducted. Much of the paper summarizes basic physical geography of tidal wetlands. That levee positions receive more sediment is not novel – in fact, its why they are levees. Similarly, that they receive more OCAR input is simply because they receive more sediment. Plant community differences similarly are not novel – the zonation of tidal wetland communities associated with deposition and salinity are well understood. What seems most interesting is the degradation aspect of the study and how these systems change with sea-level rise. I think this theme could better come forward in the paper overall.

*R7¡7 This is an interesting remark. While we do agree that the geomorphic differences between levees and basins are well known and described in literature, the difference in carbon accumulation rate between these locations is not. Therefore we tried to emphasise both the degradation aspect and the levee-basin aspect of the study, which we believe are both novel. We have altered the text in the introduction to highlight this a bit more:*

Line 109-115: "While these geomorphic differences between levees and basin are well known, it remains understudied to what extent it impacts the rate of organic carbon accumulation. . Moreover, there are currently no studies that have investigated the dynamics of OCAR along levee-basin gradients in marsh zones with a different degree of marsh degradation in response to sea level rise, which hampers our ability to predict the long-term stability of carbon in these systems as they progressively degrade in response to sea level rise.

**1.2** I am somewhat confused by the design of the sampling. As written in the text, the sampling appears quite limited. Table 1 and the text around 145-155 suggests 8 sampling sites and 4 soil cores per site. This would make n=32. However, the figures showing data points such as Figs. 2 and 6 suggest many, many more data points. The sampling needs to be clarified. Moreover, please describe how the sampling within a site is independent. Were samples collected along multiple transects? Minimum distance between soil cores? Overall how the soils were collected needs to be better described.

*R7¡8.In the initial figure 2 and figure 6 we show all measured depth intervals of every core, which indeed gives the impression of many more datapoints. Based on this comment and that of reviewer 2, we have changed our figures to show both the average value and standard deviation of each core:*

[Figure]

**Figure 1: Dry bulk density (left) and sediment accretion rates (right) determined with radiometric dating along the levee basin gradient. The coloured points indicate the average value for each core and the error bars show the standard deviation for each core. The data shown are pooled for the least degraded, intermediately degraded and most degraded zone. The letters above indicate the significance of the differences, where observations with the same letters are not significantly different from each other (derived from ANOVA for the sediment accretion and from linear mixed models for bulk density).**

[Figure]

**Figure 6: δ13C values along the degradation and levee-basin gradient. The coloured points indicate the average value for each core, the error bars indicate the standard deviation for each core.** The colours of the  points correspond to the photosynthetic pathway of the dominant vegetation (light green for C3, dark green for C4). The horizontal coloured lines correspond with the δ13C values of C3 vegetation (light green), C4 vegetation (dark green) and suspended sediment (blue). The lighter-coloured area around the lines correspond to the 95% confidence interval of the δ13C values.

*We have also changed the explanation on the sampling design to hopefully make it more clear.*

Line 163-173: "The selection of study sites resulted in eight sampling location (Table 1), two in the most degraded zone and 3 in the intermediately and most degraded zone. At each sampling location, four replicate soil cores were collected approximately one meter apart. Three replicates were used for organic carbon analysis (see 2.3.1 and 2.3.4) and one was used for radiometric to determine the sediment accretion rate (see 2.3.2). Of the total of 32 cores, 8 were used for radiometric dating and the remaining 24 for organic carbon analysis. Every core was between 25 and 50 cm long and was sliced in increments of about 1 cm. For the organic carbon analysis every other depth interval was used, leading to 12 to 25 data points for each core and a total number of between 288 and 600. "

We also reran the statistical analysis, using linear mixed model instead of ANOVA models, to account for the auto-correlation between measurements of the same cores (see R1.4).

**1.3** Similarly, I am concerned about the sampling of the gradient in degradation. The gradient is described/quantified as the ratio of vegetated to non-vegetated surface and while this ratio is reported in table 1, the sampling as I understand it is somewhat misleading since only the vegetated portions of the marsh were sampled, regardless of gradient position. Clearly the vegetated and non-vegetated portions of the marsh would experience differences in OCAR input so the decision to only sample vegetated – i.e., least degraded regardless of the degradation gradient needs to be justified and the implication of this choice clearly described.

R7¡9.*We did in fact try to sample the unvegetated parts of the marsh as well, but this turned out to be more difficult than expected and the samples were not comparable to the ones we took in the vegetated marsh. The sediment in the unvegetated parts was often very loose, making it impossible to collect solid sediment cores that could be analysed for sediment accretion rate. However, the remaining vegetated marsh is also very different along this gradient of UVVR. In the most degraded parts the vegetated marsh is more broken up into separate clumps of vegetation, compared to a continuous vegetated marsh in the least degraded sites. We added some more explanation in the methods section to explain why only the vegetated parts were sampled and how these are also different along the gradient.*

Line 151-155: " Degraded zones are characterized by destabilized vegetation zones and large pools of open water, which contain fluid mud where sampling fixed sediment volumes was not feasible. Within each zone, samples were collected on levee and on vegetated basin locations."

**1.4** The description of the statistical analysis is too limited for the statistical procedure to be evaluated. Please expand the analysis section to indicate if fixed or mixed models were used and any random effects, any data transformations, selection of post-hoc tests (the results of which are show in the figures).

R7¡0.*You are indeed correct. We changed our statistical analysis to linear mixed models, where core is used as a random effect combined with a Tukey post-hoc test to see the differences between all the sites and locations. We have changed the section on the statistical analysis as follows:*

Line 242-250: "For sediment accretion rates  the difference between levee and basin locations with *Schoenoplectus* and *Spartina* was investigated using ANOVA in R version 4.4.1 (R Core Team, 2022). For the organic carbon content, density and accumulation rate, the effect of degradation zone and location separately was investigated using linear mixed effects models, including core and depth as random factors, using the lme4 package (Bates et al., 2015). Besides the simple effect of location and degradation zone, we ran an additional model with their interaction effect. To see which locations and zones differed from each other, a Tukey post-hoc test was done using the emmeans package in R (Lenth, 2025). Bulk density was analysed in a similar way, but only looking at the difference between levee and basin locations. "

**1.5** I find several inconsistencies in the arguments surrounding the differences between the levee and basin communities. Line 380 suggests that high accretion rates in levees may be due to rapid burial of organic matter and low O2 availability leading to lower decomposition. However, on line 339, there is the suggestion that deep-soil pore water drainage on levees promotes oxygenation and more rapid plant growth. While perhaps these can both be true depending on the precise depth of anoxia, its reads as inconsistent. Similarly, on line 394, the packing of high-density mineral matter on the levees is used as a justification for the greater bulk density of levee soils would further argue against rapid drainage and oxygenation.

R7¡❶*Thank you for bringing this to our attention. We understand that the processes we're describing can be seen as inconsistent, but it is indeed the depth of the anoxia that plays a role. Since the accretion rates are high on the marsh levees, we believe that this will result in faster burial of the present carbon to layers below the oxic zone, even though this oxic zone is deeper on the levees compared to the basins. We have tried to highlight this more in the text as follows:*

Line 405-407: "Even though the sediment drainage is deeper in the levees (Armstrong et al., 1985; Balling & Resh, 1983; Mendelssohn & Seneca, 1980; Ursino et al., 2004; Van Putte et al., 2020), the observed higher accretion rate on the levees results in faster burial of the carbon present in the profile to layers below the sediment drainage level, where oxygen is less available. This could imply lower rates of decomposition and thus better preservation of the present carbon  (Rietl et al., 2021)."

**1.6** I have concerns with the interpretation of the 13C data as presented here. The sediment varies considerably in 13C suggesting different sources of OCAR input as the authors indicate. However, the endpoints of the carbon is somewhat ambiguous. The argument is made that 13C can determine the difference between autochthonous C and allochthonous C. However, autochthonous C can come from two sources – the C4 grass *Spartina* and C3 rush *Schoenoplectus* while allochthonous C is assumed to be C3 (presumably phytoplankton and other algae). Therefore, while seeing a highly C4 signature in sediment is good indication of local C in a *Spartina* zone the opposite is not necessarily true since the deposition could be from OCAR input from outside the wetland as well as OCAR input from remobilized sediment with a local source. Please address this concern in interpreting these data. Figure 6 I think shows the community shift happening with the *Schoenoplectus* OCAR being mostly C4-derived in the least degraded and intermediate sites and mostly C3 derived in the most degraded. Since this is a C3 plant, the data suggest a recent conversion (and the large error bars support this) in the least and intermediate sites but a long-term history of the C3 rush in the most degraded. Combined with the assertion that basins are sediment starved, the data argue for local carbon inputs dominating the basin system.

R7¡❷*This is a very valid concern. We have tried to formulate an additional sentence about the interpretation of the results in the C3 basin.*

Line 393-399: "...This pattern in sediment deposition is confirmed by the $\delta^{13}C$ value of the levee sediments (Fig. 6), where the average value (-21.0‰) indicates a mixture of different sources of carbon, from local C4 vegetation (-14.4‰) and incoming suspended sediment (-26.3‰). The basins under C4 vegetation in the least and intermediately degraded zones,

however, have a $\delta^{13}C$ value of (-16.2‰) that is relatively close to that of the vegetation (-14.4‰). We believe that this pattern is also valid for the basin under C3 vegetation, but that is less straightforward to determine from the $\delta^{13}C$ value, since the signature of C3 vegetation is close to that of suspended sediment. ..."

*As for figure 6 we indeed do believe that a vegetation shift has happened from Spartina dominated patches to Schoenoplectus, however it does not really fit into the story of carbon accumulation (since we observed no real differences between basin sites with Spartina or Schoenoplectus), so we decided to leave it out of this paper.*

**Response to reviewer 2**

**General Comments**

Thank you for the opportunity to review the manuscript titled "Carbon sequestration along a gradient of tidal marsh degradation in response to sea level rise" by Mona Huyzentruyt and colleagues. This paper reports on the differences in organic carbon accumulation rates among levees and basins along a marsh degradation gradient within a microtidal wetland. The results of the paper indicate that carbon accumulation rates are much greater on the levees than within the basins. Some differences in organic carbon accumulation rates were detected among the different marsh degradation zones which may suggest accumulation rates tend to be greater in more degraded zones.

*We would like to thank the reviewer for the very thorough review of our paper. We have tried to implement your comments to the best of our abilities. In particular we (1) replaced the initial statistics with linear mixed effects models, as suggested, and (2) gave more detail on the sampling methods, such as how sites were selected, how sediment samples were cored and how replicate samples were defined. Many other smaller text changes were included in response to your feedback as detailed below.*

*The original feedback is denoted in black with numbered headers (2.1 for the first comment, R2.1 for the response on the first comment), followed by our response in blue. The textual changes made in the manuscript are indicated below the response (new pieces of text in blue and removed text in* red*).*

The manuscript represents a substantial contribution to scientific progress, and important new data, in the Biogeosciences. The paper is well organized and written without errors but could be improved by editing to reduce text, and re-arranging where some information is introduced within the text.

> **2.1** There are a few issues within the statistical approach that can be addressed and will likely not have a large effect on the main results reported by the paper. Specifically, there are several analyses (dry bulk density and C-13) where it appears all of the subsamples from all cores were included as individual observations and run in an ANOVA. If this is the case then this method would artificially inflate the sample size and therefore artificially decrease the error term, as each sample is being treated as independent even though multiple samples came from the same core. To deal with this, the authors could consider a repeated measures ANOVA or linear mixed modeling approach where "depth" can be nested within "core". It could be that the authors want to consider taking this approach in other analyses as well to potentially gain a little more insight from the data they have, as it could help take advantage of all the depth data rather than just averaging it all to one value. But the other analyses are ok as is if the authors don't want to make that change.
>
> *R8¡7.This is a very valid remark. We have altered the statistical analysis to linear mixed effects models, using both depth and core as random factors. We tested the effect of degradation zone and location separately as well as their interaction.*
>
> Line 242-250: "For sediment accretion rates  the difference between levee and basin locations with *Schoenoplectus* and *Spartina* was investigated using ANOVA in R version 4.4.1 (R Core Team, 2022). For the organic carbon content, density and

accumulation rate, the effect of degradation zone and location separately was investigated using linear mixed effects models, including core and depth as random factors, using the lme4 package (Bates et al., 2015). Besides the simple effect of location and degradation zone, we ran an additional model with their interaction effect. To see which locations and zones differed from each other, a Tukey post-hoc test was done using the emmeans package in R (Lenth, 2025). Bulk density was analysed in a similar way, but only looking at the difference between levee and basin locations. "

*As for the $\delta^{13}C$ values, we did not perform any statistical testing on this data, we only show it to compare the soil carbon signatures to those of the incoming sediment and local vegetation. This gives us insight in the sources of carbon within in the system. We have clarified this in the statistical analysis paragraph.*

Line 252-253: "No statistical testing was done on the δ13C values, but they were used to compare between the soil values, the vegetation values and the suspended sediment values."

**2.2** The authors also make the claim that the degraded marsh zones are experiencing the same rate of relative sea level rise as the other zones, but this should be better discussed and documented as they also make statements that suggest the degraded zone may be decreasing in elevation (more ponds, etc) which would mean that the rate of relative sea level rise may be greater in the degraded marsh zones.

*R8¡8 This is a very good remark. This confusion is the result of an error in the original text in Line 445 where we wrote 'relative sea level rise', when we meant 'sea level rise'. Now we give a proper explanation of the sea level dynamics in this region including sea level rise, subsidence, sedimentation, and tidal reduction, etc.... We wanted to say that our entire system experiences the same local rate of sea level rise (based on the NOAA online portal at the Cambridge station: https://tidesandcurrents.noaa.gov/sltrends/), however we do agree there can be relative differences based on the marsh platform elevation. We assume that the lower elevation areas will experience a higher relative rate of sea level rise, resulting in higher inundation duration. We mainly mention this in comparison with other studies who look at different areas around the world that experience different local sea level rise rates (for example comparing marshes in the San Francisco Bay with a SLR rate of 0.92 mm/y with the Chesapeake bay). We have removed the word 'relative' in the previously mentioned paragraph and added some additional explanation:*

Line 443-449: "However, a major difference between our study and previous meta-data studies, is that our marsh degradation zones experience the same rate of local  sea level rise but show different degrees of marsh degradation in response to the sea level rise, while previous meta-data studies are based on data from different areas experiencing different rates of sea level rise. Since our sites have different elevation levels and tidal range, they likely have experience a different  sea level rise rate, but this difference will be more limited than when comparing to different systems. ..."

Additionally, we added some explanation in the materials and methods:

Line133-141: " This general accretion deficit, combined with observed subsidence (Ohenhen et al., 2023) in the system has led to severe marsh degradation. The changes in tidal range and marsh elevation (Table 1) along the river result in different relative rates of sea level rise at the different marsh locations. This has led to a spatial gradient in marsh degradation, with stable marshes close to the Fishing Bay and increasing historical conversion of marsh to ponds moving upstream along the Blackwater River (Schepers et al., 2017)."

Specific Comments:

**Graphical abstract**

- **2.3** In the graphical abstract – suggest indicating the direction of the coast is given that the setting is describing tidal marsh ecosystems.

  R8¡9 *We added arrows indicating the direction of the mainland areas and of the Chesapeake bay.*

[Figure]

**Abstract**

- **2.4** In line 24 the authors state: "Additionally, OCAR was observed to increase with increasing degree of marsh degradation in response to sea level rise" but really there is just one difference in the basin rates ('most degraded' is higher) and one difference in the levee rates ('least degraded' is lower).

*R8¡0.It is indeed correct that only one levee and one basin rate are different, however for the readability and simplicity of the abstract we have decided to keep it like this, however if you feel this is not appropriate, we will change the abstract. We have added more nuance in the discussion.*

Line 438: "This is observed both on two of the levees as well as in one of the basin locations (Fig. 5). It is however important to note that only three points along the degradation gradient were measured, so general conclusions should be made with caution. However, this result does correspond with positive relationships found between sea level rise rate and OCAR in meta-analyses based on datasets compiled from sites across continents and the globe (Herbert et al., 2021; Huyzentruyt et al., 2024; Rogers et al., 2019; Wang et al., 2019).

**Introduction**

- **2.5** Does 'marsh degradation' specifically mean less vegetation? If so, make sure this is clearly defined in the Introduction.

  *R8¡❶The most obvious sign of marsh degradation is die off of vegetation resulting in bare soil or even ponds (as specified in line 61-66). However, there are also changes in the stability of sediment (which is very noticeable while walking at these sites). Since this is mainly something we observed, it is a bit tricky to write about it in the introduction, but we did try to emphasise it in the materials and methods.*

  Line 153-156: " Degraded zones are characterized by destabilized vegetation zones and large pools of open water, which contain fluid mud where sampling fixed sediment volumes was not feasible. Within each zone, samples were collected on levee and on vegetated basin locations."

- **2.6** How wide-spread is the phenomenon of decreasing vegetation in tidal marshes globally?

  *R8¡❷Vegetation loss in tidal marshes can be caused by many processes other than sea level rise. However, the systems that we list in our introduction (line 58-60) are all systems that are experiencing losses in vegetation or surface elevation due to the inability to keep up with sea level rise, which is a globally widespread phenomenon experienced specifically in microtidal marshes (specified line 55-58). Including other sources of marsh loss, it is predicted that global wetland loss will be between 0-30% of the current area by 2100 (*Schuerch et al., 2018). *We added this global estimate to the introduction:*

  Line 60-62: "A recent global scale study has estimated that wetland (mangrove and marsh) loss will range between 0 and 30% by 2100  (Schuerch et al., 2018)."

- **2.7** How were the degradation zones (least degraded, intermediately degraded, most degraded) determined? Now I see this is answered in the supplement, but a brief explanation should be included in the paper text (the material in the supplement can remain the same).

*R8¡❸ Thank you for bringing this to our attention.. We do bring on the concept of the Unvegetated-vegetated ratio already in the introduction (line 65-67), this ratio was used to determine our degradation zones. But we have added additional clarification in the fieldwork setup paragraph:*

Line 147-151: Three marsh zones were selected along the marsh degradation gradient, based on an increasing unvegetated-vegetation ratio (UVVR, Ganju et al., 2017). These sites will further be referred to as (1) least degraded (UVVR=0), (2) intermediately degraded (UVVR=0.016) and (3) most degraded (UVVR=0.143) (Fig. 1, Fig. S1, Table 1).

- **2.8** Introduce the difference between C3 and C4 vegetation and why it matters in this context in the introduction (as they were investigated separately in this work).

    *R8¡❹ We have decided to integrate the relevance of sampling both C3 and C4 vegetation in the method section rather than the introduction. We believe it would disrupt the storyline of the introduction. We indicated the changes made to the method section in comment 2.10.*

**Methods**

- **2.9** It seems that four cores were taken within each zone, but at the same site. Why weren't cores taken from multiple sites throughout the zone... that seems like it would better represent the carbon dynamics of that zone.

    *R8¡❺ Indeed all 4 cores were taken at the same location, with at least 1 meter distance between them. This was chosen this way to limit the number of field sites and thus time needed in the field to go to each site. We do agree that more transects within one zone would be better, but there was simply not enough time.*

    Line 160: "At each sampling location four replicate cores were taken approximately one meter apart, of which one was used for radiometric dating (refer to section 2.3.2) and three were used for bulk density and organic carbon analysis."

- **2.10** Explain why two vegetation types were sampled. Perhaps the C3 vs C4 difference should be introduced in the Introduction if the authors think it is important.

    *R8¡76. We do believe that this is an important point. The first reason that two vegetation species are sampled along the gradient is because there is a clear co-dominance of these vegetation types (which is specified in the methods Line 146) along a big part of the gradient and we wanted to see whether there was a difference between these species in terms of carbon dynamics. Second, we wanted to be able to look at the source of the carbon and not only the accumulation rate. Since there is a larger difference in $\delta^{13}C$ value between C4 vegetation and incoming sediment, it makes the distinction between sources easier than with C3 vegetation. We have specified this more in the method section:*

    Line 152-156: "Because the basins of the least degraded and intermediately degraded zone contained distinct patches of two vegetation types, samples were taken within these zones at two basin locations, i.e. in each of the two vegetation types (one dominated by *S.*

*americanus,* a C3 species and the other by a mixture of *S. alterniflora* and *S. patens,* C4 species), but only one levee location was sampled (dominated by *S. cynosuroides,* a C4 species)."

and Line 223-228: "The δ¹³C values were also measured for above-ground vegetation, by analysing finely ground vegetation samples, and for suspended sediment samples. The δ¹³C signature of C3 and C4 vegetation is very different (Bouillon & Boschker, 2006; Farquhar et al., 1989), with C4 vegetation typically having a signature around -14‰, and C3 around -26‰ (Bouillon & Boschker, 2006). Since incoming sediment often has a δ¹³C signature similar to C3 vegetation, it is more straightforward to distinguish between vegetation and externally derived carbon within C4 vegetation. For the analysis of the suspended sediment samples, the filters were cut into four equal parts and one part was used for the δ¹³C analysis."

- **2.11** Why were water samples taken at just one location and how is this information used? Was the site inundated, and this was the water present above the soil surface?

  R8¡77. *These are good questions. Initially, water samples were collected in the tidal channel next to each sampling location. However due to a processing error, these could not be used for analysis. Our local contact in the US therefore went back, but did not have the time to go to each field location, so one sample was collected to at least have some data. The water sample was analysed for carbon isotopes, to have an idea of the signature of the suspended sediment and compare this to the sediment carbon isotope signatures. Since we are working in a microtidal system, tidal inundation is limited and all sediment deposition comes from the tidal creek. As stated before, the water sample was collected in the tidal creek. We have tried to clarify both the sampling and the use of the samples in the text.*

  Line 212-216: "Water samples were collected to analyse the suspended sediment for their δ¹³C value (see 2.3.4), to be able to see whether the bulk carbon was plant derived or coming from the tidal deposition. As the delivered sediment comes from the tidal channel, water samples were collected at one location  from the Blackwater River.  After collection, the samples were stored in the fridge until further analysis."

**Statistical Analyses**

- **2.12** The description of the stats leaves some questions. Were levee and basin sites from all of the degradation zones all analyzed against each other, or were levee and basin nested within zone?

  R8¡78. *In our initial analysis, we analysed them all against each other. However, in the new analysis (see below), we have changed the statistics to linear mixed models and included the effect of site and location separately as well as the interaction between both. This way we could see the overall difference in location and site as well as the specific differences between each of the combinations. (2.1 and 2.13).*

- **2.13** Figure 2. It is not statistically appropriate to pool values from different depths within one core as they are not independent. However, the authors could address this by adopting a linear mixed effects modeling framework that nests "depth" within "core". Such

an approach may actually be useful for some of the subsequent analyses as well because it may help the authors determine more about the differences in carbon density, for example, in the basin sites along the degradation gradient that currently are all the same using the ANOVA analyses, but the differences may be parsed out if all of the measurements from each core were included. It may be overkill as the authors do not specifically have 'depth' questions – but as the analyses are set up right now a lot of information (and work!!) is being 'tossed out' as the cores are averaged to just one value.

R8¡79.*You are indeed correct. We have repeated the statistical analysis with linear mixed models instead of just ANOVA testing. It has been changed in the method section (as in response R2.1).*

*And the figure captions have also been changed.*

- **2.14** Lines 266-270, as the authors point out, there is no statistical difference in the mean carbon density among the basins of different degradation zones, so that cannot be reported as a finding.

R8¡70.*You are indeed correct. We have rephased the section:*

Line 284-291: "For organic carbon densities (g cm$^{-3}$; OCD; Fig. 4) the values were significantly higher (p < 0.05) on the levees compared to the basins, for all zones along the marsh degradation gradient. There was however no significant difference between the values in the basins or levees of the different zones (Fig. 4). ~~The values in the basins also increased along the degradation gradient, with the highest basin values found in the most degraded zones (0.030 g cm$^{-3}$), followed by the intermediately degraded basin with *Schoenoplectus* (0.029 g cm$^{-3}$) and *Spartina* (0.028 g cm$^{-3}$). The lowest values were found in the least degraded basin with *Schoenoplectus* (0.025 g cm$^{-3}$) and with *Spartina* (0.024 g cm$^{-3}$).~~"

- **2.15** Introduce the C3 vs C4 difference among the two vegetation types earlier as the reasoning for separating them and explain why this is important in introduction.

R8¡7❶.*See response to comment 2.10.*

- **2.16** Why are the individual depth measurements used for the C-13 plot and not for the others, and how did the authors avoid pseudo-replication? (Does the statistical model include depth nested within core to avoid inflating the sample size and artificially shrinking the error term?) Remember that multiple depths are essentially 'repeated measures' within a core.

R8¡7❷.*We did not do any statistical testing on the δ$^{13}$C values, as they are not meant to compare between the different sites, but more to compare the soil carbon δ$^{13}$C values with the δ$^{13}$C values of the incoming sediment and the local vegetation. We have added a clarification in the method section and added the remark to the caption of figure 6.*

Line 252-253: "No statistical testing was done on the δ13C values, but they were used to compare between the soil values, the vegetation values and the suspended sediment values."

Line 322-327: "Figure 6: δ$^{13}$C values along the degradation and levee-basin gradient. The colours of the boxplots correspond to the photosynthetic pathway of the dominant

vegetation (light green for C3, dark green for C4). The horizontal coloured lines correspond with the δ¹³C values of C3 vegetation (light green), C4 vegetation (dark green) and suspended sediment (blue). The lighter-colored area around the lines correspond to the 95% confidence interval of the δ¹³C values. The dots indicate all depth interval measurements and can not be seen as individual replicates."

**Discussion**

- **2.17** Figure 7 is unnecessary.

  R8¡7❸*Thank you for this remark. We do believe that this figure adds value by placing our results in a larger global context, rather than only in a tidal marsh context. Therefore, we would like to keep this figure in the manuscript, but are of course willing to reconsider upon the insistence of the editor/reviewers.*

- **2.18** Line 312 states that the study examines: "accumulation rates (OCAR) in response to gradients in marsh degradation and levee-basin gradients." It seems there is an important distinction between examining accumulation rates across gradients of degradation, and "in response" to degradation. It seems the authors are doing the former and therefore should use that language here, i.e. change to "accumulation rates across gradients in marsh degradation."

  R8¡7❹*Thank you for noticing our error, we have changed the sentence accordingly:*

  Line 333: "In particular, knowledge is limited on sediment organic carbon accumulation rates (OCAR) along  gradients in marsh degradation and levee-basin gradients. In this study, we found that marsh levees are hotspots of OCAR,..."

- **2.19** Line 313, what is the relative area of levees to basins in this wetland, and in most tidal wetlands? This will help provide context on the relative importance of these 'hotspots'

  R8¡7❺*In our study area, the levees are between 10 and 20 m wide depending on where along the Blackwater river you are. We are currently working on a spatial study to estimate how the total carbon budget of a system is influenced by taking into account or disregarding the effect of levees. This is also mentioned in the introduction (Line 97-98) and the materials and methods (line 129). We have also added it in our discussion.*

  Line 335-337: "In this study, we found that marsh levees are hotspots of OCAR, accumulating organic carbon four times faster on average than in adjacent marsh basins. Even though there area is limited (in this case a band of 10-20m width along the river), we believe that taking the difference in carbon accumulation rate between levees and basins into account can make a big difference for system-scale carbon estimates."

- **2.20** Line 315: when the authors state that levees are "among the fastest carbon accumulating environments on Earth" they are talking specifically about soil organic carbon accumulation, right?

  R8¡86 *We do indeed mean soil organic carbon accumulation and have clarified it as such in the text.*

  Line 339: "Based on our findings, marsh levees in a micro-tidal, organogenic marsh system appear to be among the fastest **soil** carbon accumulating environments on Earth (Fig. 7)."

- **2.21** Figure 8. Clear and relatively easy to understand but at first glance the relative size of the arrows among the levee and the two vegetation communities appears to be the same. It is difficult to determine the 'point' of the conceptual figure – are there differences in the relative strength of these processes among the different locations? It seems there must be if the accumulation rates are so different, but it is difficult to see this from the figure.

    *R8¡87.We have changed the thickness of the arrows (instead of only the height) to highlight the difference in relative strength of the processes.*

[Figure]

- **2.22** Line 410: Be specific that one basin rate differs from the other two and one levee rate differs from the other two. Especially given that these rates were only sampled in one location per zone (via three cores), it seems to be overstating the results a bit to claim that there is an increase in OCAR with increasing marsh degradation.

    *R8¡88 You are correct that a bit more nuance is warranted. We have changed the sentence as follows:*

    Line 438: "This is observed both on two of the levees as well as in one of the basin locations (Fig. 5)."

- **2.23** Line 419: This is interesting - do the authors know that the degraded marsh experiences the same rate of relative sea level rise? It seems that it could be slightly different given that vegetation has been lost so perhaps rates of accretion are lower? If the degraded area is experiencing any subsidence, or even just lower rates of accretion, then it would be experiencing a faster rate of relative sea level rise.

    *R8¡89.This is a very good remark. This confusion is the result of an error in the original text in Line 445 where we wrote 'relative sea level rise', when we meant 'sea level rise'. Now we give a proper explanation of the sea level dynamics in this region including sea level rise, subsidence, sedimentation, and tidal reduction, etc.... We did not specifically measure the sea level rise rate at each location, we used the NOAA online portal for the rate of SLR ([https://tidesandcurrents.noaa.gov/sltrends/](https://tidesandcurrents.noaa.gov/sltrends/)) in our area (we used the Cambridge gauge station). We do agree that the local relative sea level rise rate in our specific locations can*

*deviate from that general value. We assume that the lower elevation areas will experience a higher relative rate of sea level rise, resulting in higher inundation duration. There is indeed evidence of subsidence in the entire Chesapeake Bay (https://www.nature.com/articles/s41467-023-37853-7) and there are indeed differences in sediment accretion, which are linked to differences in tidal range and marsh elevation.*

Line133-141: " This general accretion deficit, combined with observed subsidence (Ohenhen et al., 2023) in the system has led to severe marsh degradation. The changes in tidal range and marsh elevation (Table 1) along the river result in different relative rates of sea level rise at the different marsh locations. This has led to a spatial gradient in marsh degradation, with stable marshes close to the Fishing Bay and increasing historical conversion of marsh to ponds moving upstream along the Blackwater River (Schepers et al., 2017)."

Line 443-450: "However, a major difference between our study and previous meta-data studies, is that our marsh degradation zones experience the same rate of local  sea level rise but show different degrees of marsh degradation in response to the sea level rise, while previous meta-data studies are based on data from different areas experiencing different rates of sea level rise. Since our sites have different elevation levels and tidal range, they likely have experience a slightly different relative sea level rise rate, but this difference will be more limited than when comparing to different systems. Hence an alternative explanation must be sought for the results in the Blackwater marshes."

- **2.24** Lines 430-435: all of the processes described between Line 430 and 435 indicate that degraded marshes do experience sediment loss which would then make them vulnerable to higher rates of relative sea level rise.

  *R8¡8❶Yes, you are right. As detailed in R2.23 above, we made an error in the original text at line 445, where 'relative' sea level rise was written when we meant absolute 'sea level rise', as measured in this regions via publicly available NOAA portal (https://tidesandcurrents.noaa.gov/sltrends/). We hope the elaborated discussion and text changes in R2.23 resolve this point of confusion.*

**Technical Corrections:**

- **2.25** Section 4.1.1. Minor writing suggestion - three sentences in a row start with "This", consider rephrasing to reduce redundancy. The section could likely also be condensed.

  *R8¡8❶Thank you for noticing. We have changed the paragraph as follows:*

Line 367-371: "This is a consequence of facilitated pore water drainage towards creeks that are located next to levees, while pore water drainage from basins is hindered as they are much farther away from creeks (Armstrong et al., 1985; Balling & Resh, 1983; Mendelssohn & Seneca, 1980; Ursino et al., 2004; Van Putte et al., 2020).  The deeper drainage on levees leads to better soil aeration during low tides (Mendelssohn & Seneca, 1980) and thus better conditions for vegetation growth (Callaway et al., 1997; Kirby & Gosselink, 1976).  Other studies have found a similar pattern for multiple species, such as *Salicornia* (Balling & Resh, 1983) and *Spartina alterniflora* (Kirby & Gosselink, 1976)."

- **2.26** Line 355: remove "be expected to"

  *R8¡8❷We removed be expected to.*

- **2.27** Line 368: remove one parenthesis after "Ganju et al., 2013"

  R8¡8❸*Thank you for noticing, we removed the extra parenthesis.*

---

## Author Response (AR2)

**Response to reviewer**

Additional private note (visible to authors and reviewers only): The manuscript has now been seen by a 3rd reviewer, who has recommended it for publication pending the following corrections.

*We would like to thank the additional reviewer for their comments on the manuscript. The response is structured as follows, every comment got a number, which is matched by the number of the response. Text that is added in the manuscript is shown here in* blue, *text that is removed is shown in*  *and all things in the text that have changed are indicated in the boxes underneath the response.*

1. In the graphical abstract, "gradient in marsh degradation" with an arrow doesn't necessary suggest to me if the increase is away or toward the bay.
   *R1: We changed the graphical abstract to include the unvegetated-vegetated ratio (UVVR). Additionally, we moved the arrows 'towards the mainland' and 'towards the inland' to the other side, so that it is linked with the arrow of degradation.*

   **Line 34:**

[Figure]

2. Further, I'm not sure if degradation in general is a concept that is clear to most readers as you've defined it.
   As someone not familiar with your study system, it took me until Ln 361 to know how degradation was defined in the study. I would put this information in the introduction, with the first presentation of the term.
   *R2: We tried to define it clearly in line 64-70, but we tried to make it even more clear.*

   > **Line 64-71**: The resulting bare soil patches or shallow ponds that form inside marshes, and their surface area relative to the surrounding remaining vegetated marsh area (so-called unvegetated-vegetated ratio, UVVR), is considered here a proxy for the state or degree of marsh degradation (with higher UVVR indicating a higher degree of degradation), in line with previous studies (Ganju et al., 2017).

> An important question is how this degree of marsh degradation (measured as UVVR) in response to sea level rise affects the organic carbon sequestration efficiency in the remaining vegetated marsh zones.

3. Ln 50-55 Repeat of word choice makes it hard to read.
   *R3: The feedback is indeed mentioned twice, so we tried to make it more readable by changing the sentence.*

   > **Line 49-55:** On the one hand certain marshes can keep up with sea level rise, due to positive feedbacks between tidal inundation duration, sediment accretion, and surface elevation gain, in particular macro-tidal marshes with high sediment supply (Kirwan et al., 2016). For such marsh sites previous studies have found an increase in organic carbon accumulation rate with increasing sea level rise rate, due to the earlier mentioned positive feedback  organic carbon accumulation rate (Herbert et al., 2021; Huyzentruyt et al., 2024; Suello et al., 2025; Wang et al., 2021).

4. Figure 1. I am assuming the orange square on the map is the whole map underneath – however, it seems that the bottom should have the Chesapeake Bay and it does not. I don't see the two lakes in the satellite image in the square. I assume the least degraded is closer to Chesapeake Bay since you say the Fishing Bay is a tributary, but I can't see that from the map shown.
   *R4: We included an additional arrow on the map indicating the direction of the Chesapeake Bay.*

   **Line 143:**

[Figure]

5. Ln 354 – I would be careful with a statement like this. It is always hard to know something is the fastest. I think it's good you tempered with "among the fastest" maybe also temper with "of known rates" since of course many are not measured.
   *R5: We added it to the sentence.*

**Line 360:** Based on our findings, marsh levees in a micro-tidal, organogenic marsh system appear to be among the fastest soil carbon accumulating environments on Earth of known rates (Fig. 7).

6. Minor comments:
   Ln 179: 25 cm x 25 cm
   Ln 180: laboratory not lab
   Ln 350: their not there
   *R6: We made these small changes.*

7. Figure – I'm not a fan of the lines across the figures or the Letters for significance so far above the data.
   *R7: We have lowered the letters of significance, however we believe that the horizontal lines do improve the readability of the figures, so we decided to keep it like this. We give one example of a changed figure here, but we changed all of them.*

**Line 305:**

[Figure]

8. Figure 7 legend should have superscripts for units
   *R8: Thank you for noticing, we made the units into superscripts.*

**Line 371-374:**

**Figure 7: Overview of the modern-day carbon sequestration rates (expressed in g C m$^{-2}$ y$^{-1}$) in different ecosystems (adjusted from Temmink et al., 2022), including indications of the average rates measured on our levee and basin locations. Error bars indicate the standard deviation of measurements.**